# A new region-aware bias correction method for simulated precipitation in areas of complex orography

Juan José Gómez-Navarro[1,2,3], Christoph C. Raible[1,2], Denica Bozhinova[1,2], Olivia Martius[2,4], Juan Andrés García Valero[3,5], and Juan Pedro Montávez[3]

[1]Climate and Environmental Physics, University of Bern, Bern, Switzerland
[2]Oeschger Centre for Climate Change Research, Bern, Switzerland
[3]Department of Physics, University of Murcia, Murcia, Spain
[4]Institute of Geography, University of Bern, Bern, Switzerland
[5]AEMET, Agencia Estatal de Meteorología, Spain

**Correspondence:** Juan José Gómez-Navarro (jjgomeznavarro@um.es)

**Abstract.** Regional climate modelling is used to simulate the hydrological cycle, which is fundamental for climate impact investigations. However, the output of these models is affected by biases that hamper its direct use in impact modelling. Here, we present two high-resolution (2 km) climate simulations of precipitation in the Alpine region, evaluate their performance over Switzerland, and develop a new bias correction technique for precipitation suitable for complex topography. The latter is based on quantile mapping, which is applied separately across a number of non-overlapping regions defined through cluster analysis. This technique allows removing prominent biases while it aims at minimising the disturbances to the physical consistency inherent in all statistical corrections of simulated data.

The simulations span the period 1979-2005 and are carried out with the Weather Research and Forecasting model (WRF), driven by the reanalysis ERA-Interim (hereafter WRF-ERA), and the Community Earth System Model (hereafter WRF-CESM). The simulated precipitation is in both cases validated against observations in Switzerland. In a first step, the area is classified into regions of similar temporal variability of precipitation. Similar spatial patterns emerge in all datasets, with a clear Northwest-Southeast separation following the main orographic features of this region. The daily evolution and the annual cycle of precipitation in WRF-ERA closely reproduces the observations. Conversely, WRF-CESM shows a different seasonality with peak precipitation in Winter and not in Summer as in the observations or in WRF-ERA. The application of the new bias correction technique minimises systematic biases in the WRF-CESM simulation, and substantially improves the seasonality, while the temporal and physical consistency of simulated precipitation is greatly preserved.

## 1 Introduction

Producing reliable climate information is fundamental to address many of the currently open research questions about climate change (IPCC, 2013). Many of these questions pertain the future evolution of hydrological variables, as they are especially important for potentially impacting society. An important source of uncertainty in current climate projections originates from the inability to resolve all relevant processes of the hydrological cycle, e.g. convection, which affect in particular statements

about extreme events of hydrological variables (IPCC-SREX, 2012). For instance, Rajczak et al. (2013) used simulations from the ENSEMBLE project to conclude that in the Alpine region some simulations project an intensification of heavy precipitation events during fall, albeit this result is clearly model-dependent. More recently, Rajczak and Schär (2017) updated this results using an large ensemble of 100 Regional Climate Model (RCM) simulations from both ENSEMBLES and EURO-CORDEX. These authors indicate that newer simulations exhibit no clear agreement on the projection of a reduction in summer precipitation and rainy days, and point out to the use of different convection parametrizations as one of the main sources of this uncertainty. In this regard, Giorgi et al. (2016) have shown how convective precipitation is indeed a fundamental mechanism that modulates the response of precipitation in the Alpine region to climate change.

To gain insights in the hydrological cycle, different sources of information are available, namely observations and model simulations. Particularly important for this study are gridded observational products (e.g. Haylock et al., 2008; MeteoSwiss, 2016), as their spatial homogeneity becomes particularly useful in the validation of climate models (Gómez-Navarro et al., 2012). Simulation of the climate is performed with a wide variety of models ranging from simple box models to state-of-the-art comprehensive Earth System Models (ESM) (e.g. Hurrell et al., 2013; Lehner et al., 2015). These models are used in, e.g., process understanding as well as in simulating past, present, and future climate conditions. Observations and simulations offer complementary viewpoints to climate variability. The cornerstone of climate simulations is their internal physical consistency, which emerges from the underlying set of physical equations that are solved internally as part of the simulation. However, internal variability, the counterpart of natural variability in the model world, precludes the simulation from following the actual path of climate, which indeed can be seen as a single random realization of such variability. As a compromise between models and observations, reanalysis products combine the physical consistency of climate simulations with the assimilation of observations, therefore blending physical consistency with a temporal evolution that mimics the actual past evolution of climate (e.g. Dee et al., 2011). Both ESMs and reanalysis products are useful in different contexts, and the choice of using one over the other depends ultimately on the question being addressed.

Regardless of the type of simulation being employed, a bottleneck is the spatial resolution. Global reanalysis products or simulations with state-of-the-art ESMs, e.g. in Climate Model Intercomparison Project (CMIP5) (Taylor et al., 2012; Wang et al., 2014), have a spatial resolution of 50 to 200 km (Dee et al., 2011; Rienecker et al., 2011; Taylor et al., 2012; Lehner et al., 2015). Although this spatial resolution is sufficient to explicitly simulate the physical processes that dominate the large-scale atmospheric dynamics, it cannot resolve the sub-grid physical processes that are important for the hydrological cycle, e.g., microphysics and convective processes, and therefore have to be parametrized, thereby being an important source of uncertainty in current climate projections (Rajczak and Schär, 2017). This is especially problematic for the accurate simulation of the climate in areas of complex topography, such as the Alps (Rajczak et al., 2013; Torma et al., 2015; Giorgi et al., 2016; Rajczak and Schär, 2017, among others), and in variables for which the interaction with terrain is very important, such as precipitation and wind (Montesarchio et al., 2014; Gómez-Navarro et al., 2015).

One way to overcome these problems is to increase the spatial resolution enabling the explicit simulation of a wider range of physical phenomena over the area of interest with help of a RCM. This so-called dynamical downscaling approach allows to simulate the climate over a limited-area domain according to the initial and boundary conditions prescribed by either a ESM

or a reanalysis product (Jacob et al., 2013; Rajczak et al., 2013; Kotlarski et al., 2014; Torma et al., 2015; Fantini et al., 2016; Giorgi et al., 2016, among others). The use of RCMs has proven to be a very valuable tool to downscale global datasets in the Alpine region, and indeed it has been the target area of various studies under the umbrella of large coordinate projects such as ENSEMBLES and more recently EURO-CORDEX and MED-CORDEX (e.g. Torma et al., 2015; Casanueva et al.,

2016; Giorgi et al., 2016). For wind, Gómez-Navarro et al. (2015) proved that a change in spatial resolution from 6 km to 2 km has a great impact in the ability of the simulation to reproduce the observed surface wind. Regarding hydrological variables, several studies within the frame of EURO-CORDEX have recently evaluated the added value of increasing the RCM resolution from 0.44º to 0.11º in the spatial patterns and daily variability of precipitation (Torma et al., 2015; Casanueva et al., 2016; Fantini et al., 2016; Giorgi et al., 2016). At even higher spatial resolution, Ban et al. (2014) showed that an increase in

horizontal resolution from 12 km to 2.2 km leads to a noticeably increased ability of the same model configuration to simulate the observed frequency of heavy hourly precipitation events. This improvement with increasing resolution has been confirmed using a different RCM in a similar area of study (Montesarchio et al., 2014). The reason for this improvement is that convective precipitation is explicitly simulated, which otherwise has to be parametrized being a major source of model uncertainties (Awan et al., 2011).

So far, regional simulations performed with different RCMs over complex terrain with resolutions from 2 to 25 km have been analyzed. Rajczak et al. (2013) used 10 RCM simulations for the Alpine region in the context of the ENSEMBLES project, where the horizontal resolution was set to 25 km. The conclusions drawn in the former study were validated and updated using a 100-member ensemble which includes the former runs plus the newer EURO-CORDEX simulations, in which the spatial resolution is set to 12 km (Rajczak and Schär, 2017). A number of recent studies have further improved the spatial

resolution. Montesarchio et al. (2014) conducted a simulation with the COSMO-CLM for the period 1979-2000 driven by ERA-40 reanalysis at a spatial resolution of about 8 km. This simulation allows for a satisfactory representation of temperature and precipitation, and clearly outperforms a simulation run with the same model setup but at a coarser resolution of 25 km. Ban et al. (2014) carried out a similar simulation also with COSMO-CLM for the 10-year period 1998-2007 driven with ERA-Interim with an increased resolution of 2.2 km, therefore being able to explicitly simulate convection processes.

Still, noticeable and systematic biases remain that can be attributed to either limited process understanding, insufficient reso-lution, or biases introduced by the driving dataset (Themeßl et al., 2011). To overcome this, statistical post-processing of RCM output is used to remove known systematic biases (Gudmundsson et al., 2012; Teutschbein and Seibert, 2012; Maraun, 2016). The underlying idea is to apply a statistical transformation to the simulated model output so that the distribution of modelled data resembles the observed one. There are a variety of correction methods, which can be broadly classified into distribution

derived transformations, parametric transformations and nonparametric transformations (Gudmundsson et al., 2012). Various studies have reviewed the possibilities, with an overall emphasis on hydrological variables, and quantile mapping has emerged as a nonparametric method that slightly outperforms other approaches, at least in areas of complex topography (Themeßl et al., 2011; Gudmundsson et al., 2012; Teutschbein and Seibert, 2012). Different versions of these techniques have been tested in the recent literature, and even software packages have been specifically developed and made publicly available, e.g. downscaleR

(https://github.com/SantanderMetGroup/downscaleR). Casanueva et al. (2016) applied three different methodologies to correct

daily precipitation within the EURO-CORDEX ensemble and found that the improvements introduced by the correction depends on the model, region and details of the methodology, concluding that there is no single optimal approach. Dosio (2016) used the same RCM ensemble to produce an ensemble of bias-corrected projections of climate change based on a number of climate indices from the Expert Team on Climate Change. The authors conclude that results depend on the index, season and region of interest. In particular, percentile-based indices are barely affected by bias adjustment, whereas absolute-threshold indices are very sensitive to the techniques. Further, some refinements to these techniques have been proposed. Wetterhall et al. (2012) proposed to correct the model output differently for each day, conditioned to several types of circulation patterns. Argüeso et al. (2013) introduced a variant of quantile mapping that is not corrected against gridded observations, but station data. This allows to overcome an emerging problem in very high-resolution simulations, namely that they produce fewer rain days than gridded observations, which is an assumption most bias correction techniques are based on. Felder et al. (2018) applied a preliminarily bias-corrected version of the dataset of simulated precipitation we thoughtfully present here as part of a larger study aimed at the simulation of impacts of extreme events with a compressive model chain. It this study, the authors apply and briefly evaluate a simple bias correction method, where some limitations of the technique, imposed by the complexity of the Alpine region and the high resolution of the data set, stand out. Indeed, the latter study motivated some of the improvements to the bias correction we introduce and analyse in the present study.

Despite the abundant literature on the suitability and added value of these techniques, the use of bias correction is still intensely debated. Maraun (2016) argues that it is difficult to establish the actual performance of these techniques in climate simulations, and Maraun et al. (2017) demonstrates how statistical corrections cannot overcome fundamental deficiencies in climate models, pointing out that new process-informed methods should be developed. These limitations have implications in studies addressing climate change and impacts, as the climate change signal can be unrealistically yet unwittingly modified (see discussion in Teng et al., 2015; Casanueva et al., 2018). These concerns are acknowledged and summarised in a report from the IPCC (Stocker et al., 2015). Among other recommendations, this report advices to identify and try to understand most prominent model deficiencies prior applying any bias corrections, as well as always proving the raw uncorrected data along with a clear description of the methodology applied to remove biases. In this direction, a new initiative associated to the CORDEX experiment called Bias Correction Intercomparison Project (BCIP, Nikulin et al., 2015) has been created and aims to "i) quantify what level of uncertainties bias adjustment introduces to workflow of climate information, ii) advance bias-adjustment technique and iii) provide the best practice on use of the bias-adjusted climate simulations".

Here, we tackle some of the problems discussed by Maraun et al. (2017), and demonstrated in practice in the low performance of a preliminary bias correction dataset of precipitation in the Aare catchment by Felder et al. (2018). We describe an improved approach based on the combination of dynamical downscaling to a very high resolution that explicitly considers a greater number of physical processes at regional scale, followed by a quantile mapping correction applied separately to regions which are defined according to their different precipitation regimes. Thus, the aim of this study is twofold. First, we describe two high-resolution climate simulations (2 km horizontal resolution) for the Alpine region in the period 1979-2005, and assess their performance over Switzerland with the emphasis put on the ability of the model to reproduce precipitation. These simulations supersedes existing studies (Ban et al., 2014; Montesarchio et al., 2014) in terms of length (27 years) and spatial resolution (2

km). The RCM is driven by two different datasets: the reanalysis ERA-Interim (Dee et al., 2011) and a transient simulation of an ESM (Lehner et al., 2015). The comparison of both datasets allows the characterization of errors and their attribution to biases in the driving conditions, therefore fulfilling recommendations by the IPCC for the AR6 (Stocker et al., 2015), while it enables the identification of robust features, which increases the reliability of both simulations. Second, the new process-informed bias correction technique for precipitation is introduced and applied to the simulation driven by the ESM. Thereby we can evaluate improvements with respect to previous results obtained with more simple bias correction techniques that do not explicitly account for complex topography (Felder et al., 2018).

## 2   Data, model and experimental design

### 2.1   Gridded observational dataset

This study relies on an observational dataset to evaluate and bias-correct precipitation in our model simulations. We use the gridded product RhiresD, developed by MeteoSwiss (2016). This product is based on daily precipitation totals as recorded by a network of rain-gauge stations of MeteoSwiss. It uses quality checked observations to ensure maximum effective resolution and accuracy. The observations undergo an interpolation to fill a homogeneous 1 by 1 km grid with an effective resolution of 15 to 20 km. To directly compare the observations to the simulations, we bi-linearly interpolated the observations to 2 km. Although this dataset is considered as generally reliable, it may underestimate precipitation in high altitudes due to the data sparsity (e.g., Messmer et al., 2017). More generally, observational products contain uncertainty whose magnitude can be sometimes comparable to model errors (Gómez-Navarro et al., 2012). Still, in this study we do not explicitly consider this uncertainty, and instead assume that these observations represent the true precipitation without errors.

### 2.2   Global Reanalysis: ERA-Interim

The reanalysis ERA-Interim (Dee et al., 2011) is used to provide boundary conditions for one of the RCM simulations. ERA-Interim is a reanalysis product released by the European Centre for Medium Range Weather Forecast, and is generated running the IFS model at a spectral resolution of T255 and 60 vertical levels while it assimilates observational data. The assimilation technique is the 4-D variational analysis that digest a number of observations of the actual state of the atmosphere (Dee et al., 2011). While the reanalysis covers the period from 1979 to today, a shorter period spanning 1979–2005 is downscaled. The reanalysis data used has a 6-hourly temporal resolution and a spatial resolution of $0.75° \times 0.75°$.

### 2.3   Global model simulation: CESM

The second dataset which provides boundary conditioned of the RCM simulations is obtained from a seamless transient simulation with the Community Earth System Model (CESM, 1.0.1 release; Hurrell et al., 2013). This model is a state-of-the-art fully-coupled Earth System Model developed by the National Center for Atmospheric Research run at a resolution of about $1°$ in all physical model components (atmosphere, ocean, land and sea ice) (CCSM; Gent et al., 2011) and the carbon cycle

module. The latter interactively calculates $CO_2$ concentrations and exchange these between the model components. Further details for the particular setting are presented in Lehner et al. (2015).

The transient simulation spans the entire last millennium from AD 850 to 2099, but for this study we focus on the period 1979 to 2005. The simulation is initialized from a 500-yr control simulation under perpetual AD 850 conditions. The transient external forcing is obtained from the Paleo Model Intercomparison Project 3 (PMIP3) protocols (Schmidt et al., 2011). It consists of Total Solar Irradiance (TSI), volcanic and anthropogenic aerosols, land use change, and greenhouse gases. TSI forcing deviates from the PMIP3 protocol, as the amplitude between the Maunder Minimum (1640-1715) and today is doubled. Note further that $CO_2$ concentrations obtained by the carbon cycle module are radiatively inactive. Instead, observed/reconstructed $CO_2$ concentrations (according to the PMIP3 protocol) are applied in the radiation schemes of the physical model components. Beyond AD 2005 the external forcing is obtained from the Representative Concentration Pathways RCP8.5, which corresponds to a radiative forcing of approximately 8.5 W m$^{-2}$ in the year 2100. Further details on the simulation are summarized in Lehner et al. (2015) and analyses of this simulation are presented elsewhere (Keller et al., 2015; PAGES 2k-PMIP3 group, 2015; Camenisch et al., 2016; Chikamoto et al., 2016).

## 2.4 The regional climate model WRF

The dynamical downscaling of the reanalysis data and the CESM simulation is performed with the Weather Research and Forecasting Model (WRF, version 3.5; Skamarock et al., 2008). This non-hydrostatic model uses a Eulerian mass-coordinate solver. The setting follows the one discussed in Gómez-Navarro et al. (2015): It is vertically discretized by a terrain-following eta-coordinate system with 40 levels. Horizontally, we use four two-way nested domains with grid sizes of 54, 18, 6 and 2 km, respectively (top map in Fig. 1). Although the innermost domain of the simulation spans the Alpine region almost entirely, the analysis hereafter is based on the area covered by RhiresD, which is limited to the interior of Switzerland (bottom map in Fig. 1). The physical parametrizations include the micro-physics WRF single-moment six-class scheme (Hong and Lim, 2006), the Kain-Fritsch scheme for cumulus parametrization (Kain, 2004), which is implemented only in the two outermost domains. In the inner most domain the convection parametrization is disabled as at this resolution the model is convection-permitting. The planetary boundary layer is parametrized by a modified version of the fully non-local scheme developed at Yonsei University (hereafter YSU) (Hong et al., 2006), which accounts for unresolved orography (Jiménez and Dudhia, 2012). The radiation is treated by the Rapid and accurate Radiative Transfer Model (RRTM) (Mlawer et al., 1997) and the short-wave radiation scheme by (Dudhia, 1989). Finally, land processes are simulated by the Noah land soil model Chen and Dudhia (2001).

## 2.5 Experimental design: downscaling ERA-Interim and CESM

Two RCM simulations for the European Alps are conducted for the same period 1979-2005. This period is chosen for being the overlap between the ERA-Interim and the CESM simulation. First, the ERA-Interim reanalysis dataset is dynamically downscaled with WRF (hereinafter referred as WRF-ERA). The simulation is run in so-called reforecast mode. This consists of dividing the full period into small tranches of 6 days with a spin-up period of 12 hours. This approach allows to efficiently parallelize the problem, although it has the drawback of reducing the coupling between the land and the atmosphere. This can,

in turn, introduce biases in the simulation of phenomena where the feedback between both systems is of prominent relevance, e.g. severe drought or certain type of floodings. Still, it does not impose a bottleneck of the model performance in terms of its ability to simulate surface wind, as shown by Gómez-Navarro et al. (2015), or in precipitation, as demonstrated here. Further, analysis nudging of wind, temperature and humidity above the PBL is allowed within the regional model domain, as this setting proved to outperform other configurations for this domain and model setup (Gómez-Navarro et al., 2015).

Secondly, this period of the CESM simulation is dynamically downscaled (hereinafter referred as WRF-CESM). For this simulation, the WRF setup is almost identical to the one of WRF-ERA in order to facilitate comparison between the simulations and to be able to analyze the influence of different driving datasets. Still, one important difference exists: the absence of analysis nudging. The rationale behind this choice is that avoiding nudging gives the model more freedom to develop a more precise representation of the physical processes at regional scales (due to the higher resolution), and thus is potentially able to better correct systematic biases of the ESM, which, e.g., simulate a too strong zonal circulation (Bracegirdle et al., 2013).

The comparison between WRF-ERA and WRF-CESM allows the identification of biases attributable to the driving conditions for the RCM, as described below. In this regard, it would be desirable to repeat the latter simulation using different Global Climate Models. Unfortunately the high resolution used in the RCM configuration demands a high computational cost that currently precludes the repetition of the experiment to produce an ensemble.

## 3   Bias correction technique

Although dynamical downscaling should improve coarsely resolved datasets, biases from either the driving dataset or the regional model still remain, as shown in the next section. In a previous study, Felder et al. (2018) used a bias-corrected version of the precipitation in WRF-CESM. The results (see Figs. 4 and 5 in Felder et al., 2018) demonstrate a modest performance of quantile mapping, and motivate further improvements to the methodology. Therefore, we developed a new bias correction technique, which combines a cluster analysis-based selection of regions with similar variability and quantile mapping for these regions. This technique is applied to each month separately which is justified as biases can be related to processes which undergo a strong seasonal cycle. This separation into regions of similar variability and through the annual cycle explicitly acknowledges that errors can be due to different physical processes, and therefore allows more physically coherent corrections.

In the first step, regions of similar variability are defined according to an objective criterion. In doing so, an Empirical Orthogonal Functions (EOF) analysis is applied to the precipitation series in order to obtain a rank-reduced phase space where the search of distances necessary in the subsequent cluster analysis is facilitated. We retain 7 leading EOFs, as they account for more than 80% of the total variance in the original datasets, while drastically reduces the computational cost. Then, a hierarchical clustering approach identifies regions of similar precipitation variability in the rank-reduced EOF space according to the Ward algorithm (Ward, 1963). To minimise the inherent subjectivity in the choice of the number of clusters to retain, we use a method based on the spectra of distances after every merge. To find the number of cluster centroids, the Euclidian distances between the centroids need to show a noticeable gap in the dendrogram that is built as part of the clustering procedure (not shown). A complementary criterion consists of aiming at retaining a low number of cluster centroids (and thus regions)

so that a large number of grid points per centroid is available, which will improve the estimation of the transfer function in the quantile mapping step. The resulting cluster centroids are then used as initial seeds for a $k$-means clustering, which allows for fine-rearranging of grid points across regions (as one drawback of the hierarchical clustering is that a grid point once attributed to a specific cluster centroid will belong to it despite the fact that it might be more meaningfully attached to another cluster

centroid in the end). Note that this regionalisation is not only a preliminary step of the bias correction procedure, but it is also used as an analysis technique to investigate the variability of precipitation over Switzerland and how consistent it is through various datasets.

In the second step, quantile mapping is applied separately to each of the regions identified within the first step. This non-parametric method corrects the empirical cumulative distribution function (ECDF) of the simulated precipitation with the

observation (Themeßl et al., 2011; Rajczak et al., 2016). Assume that the climate model daily time series is $X_{\text{model}}(t,x,y)$ with $t$ the time and $x,y$ the location. To obtain a corrected time series $X_{\text{corr}}(t,x,y)$ the following rule is used:

$$X_{\text{corr}}(t,x,y) = \left(\text{ECDF}_{\text{obs}}(t,x,y)\right)^{-1}\left(\text{ECDF}_{\text{model}}(t,x,y)\right)$$

with $\text{ECDF}^{-1}$ indicating the inverse ECDF, i.e., a quantile. Therefore, it can be seen as a transfer function between the ECDFs of the simulation and the observations. The quantile interval is set to 1, so quantiles corresponding to percentiles from 1st to

the 99th are corrected. The transfer function is obtained for each region independently by pooling all grid points that belong to it (therefore a larger number of grid points per cluster facilitates the estimation of such function, as outlined above). Finally, the correction is applied to the daily series of precipitation in every grid point, with a transfer function that is common to all elements within the same region, but varies across the various regions defined by the cluster analysis. A small drawback of the separation into regions is that they lead to artificial and abrupt boundaries across the domain that would leave a fingerprint

in the corrected data. To minimise these artificial boundaries, we perform a spatial smoothing in the obtained quantiles with a radius of 4 km, which smooths out the transfer functions prior to the correction, effectively removing such artifacts. Note that this scheme can lead to wet biases after the correction when the dry-day frequency is underestimated by the model, which then become systematically mapped onto precipitation days. These biases can be further removed with frequency adaptation techniques (Themeßl et al., 2012), although we do not consider them in our scheme, which can be related to wet biases in the

corrected precipitation in Winter (see discussion below).

It is important to note the rationale for the separation into regions. Quantile mapping can be in principle used either for each grid point separately or on the entire domain, here Switzerland. Both options have advantages and disadvantages. Using an average transfer function over a large heterogeneous region may lead to problems when it contains positive and negative biases that can cancel each other and disable any correction. This problem disappears applying a correction to each grid point

separately, but it has the disadvantage that the potential gain of a highly resolved physical consistent estimate of the climate obtained by the regional model is destroyed. These caveats contribute to the on-going discussion on the suitability of bias correction techniques and the necessity of more physical-based methods (Maraun et al., 2017). In this sense, the new bias correction technique based on objective regionalization presents a compromise between these two extremes, as regions with

similar precipitation behavior are corrected coherent and jointly, thus preserving a great part of the physical self-consistency of this variable for each region dictated by the RCM, but still avoiding the cancellation of positive and negative biases.

We note that the application of this methodology implies a previous regionalisation of the series for each month separately, which in general involves notable computational cost. Further, months belonging to the same season behave similarly, so that the resulting regions are hardly distinguishable and the analysis presents some level of redundancy. For these reasons, we propose a simplified form of the methodology, which we apply hereafter, and consists of carrying out the regionalisation on a seasonal basis. Once identified, these regions can be regarded as representative and common for the three months within each season, so that the final correction can be applied on a monthly basis.

## 4 Evaluation of the simulations

### 4.1 Regions of common variability and time behavior

Using the cluster analysis introduced in Sec. 3, the number of regions with common variability (clusters) slightly varies per season and dataset (Table 1). Their spatial distribution is depicted in Fig 2, where different colors represent grid points belonging to each region, and the number of grid points within the Swiss domain that belong each region is shown in Table 2. Note that in the smallest region the number of grid points is 60, which implies that 48600 pairs of numbers (i.e. 27 years × 30 days per year × 60 points per day) are used to obtain the transfer function that effectively carries out the correction in the less favourable case. This ensures that such function is efficiently estimated from the sample in all regions and cases. The number of clusters obtained is similar in all cases, and a clear Northwest-Southeast pattern emerges concurrently with the main orographic features over Switzerland (see bottom of Fig. 1). The resemblance between the regions obtained for both WRF simulations is remarkable. In all cases, a large region that includes the plains in the center of Switzerland, but also the Valais and Engadin valleys, stands out. Further, the southern part of the country, South of the Alps also emerges as a distinct region, although in some cases it is further sub-divided (see SON in the WRF-ERA simulation). The Alps themselves are another cluster in most of the seasons and datasets. The orographic pattern is explicit, with a cluster encompassing the mountains tops, in Winter in both simulations, and Spring in the WRF-CESM case. Such strong differentiation as a function of terrain height is not so explicit in other seasons. Still, it should be noted that differences in the sub-regions beyond North and South of the Alps are not so robust, and might be attributed to the subjective component in the choice of number of regions. The similarity between the regions in both simulations indicates that the precipitation regimes across Switzerland are mostly imposed by the RCM, being robust regarding the boundaries that impose the temporal evolution of the simulation. This is a non-trivial finding, as the CESM simulation is affected by acknowledged biases compared to ERA-Interim, and thus the output of the regionalization might shed to very different results. Instead, and although such biases leave a strong footprint in the amount and location of the simulated precipitation (further discussed below), the CESM boundary conditions lead to a spatial distribution of precipitation variability that, once dynamically downscaled, is greatly consistent with ERA-Interim.

Larger differences appear however when comparing the regions obtained with both simulations to the observations. As in the case of the simulations, two main superclusters stand out covering both sides of the Alps through the annual cycle, with

some seasonal differences (the Northwest-Southeast pattern is less dominant in Autumn and Winter). The presence of the Alps and its orographic footprint is less obvious, and the regions are defined with clear boundaries. There are a number of reasons that help to explain such differences. The most prominent is the different resolution. OBS has an effective resolution of about 20 km (see 2.1), whereas both simulations reach 2 km in the innermost domain (although the regionalization has been obtained with a coarser resolution version of the data of 6 km due to computational constrains). Note that the effective resolution of the simulations is coarser than 2 km, as it is between 2 and 4 times the one implemented in the simulation (Pielke Sr, 2013). The coarser resolution in the gridded product of observations contributes to the smoothing of the regions and therefore to their clearer definition. The absence of strong orographic features (mountain tops, valleys, etc.) that can be recognized in Fig. 2 for the gridded observations might be attributable to the combined effect of coarser effective resolution plus the fact that there are fewer observations in the high mountain regions. This is an important limiting factor in gridded products for precipitation in complex topography areas.

The rationale of regionalization consists of finding groups of grid points where precipitation variability within such region is coherent, whereas differences between different regions are maximized. The discussion so far has focussed on a qualitative description of the outcome of the regionalisation, without analyzing in detail to what extent these regions can be regarded as different (the dendrograms used to stablish the number of regions are not shown, for instance). Therefore we analyze next in a quantitative fashion the coherence of the regions through correlation analysis. For this, the daily precipitation series in each grid point is grouped for each region and averaged to obtain regional series. Then the cross-correlation between all series is calculated for each dataset and season, and shown with a color scale in Fig. 3. Note that there is no one-to-one correspondence between the regions in different datasets and seasons, so the labeling (1 to 6) of this figures has to be carefully read from Fig. 2. Correlations of daily regional-averaged precipitation are generally large, above 0.7 in many cases and never negative. This indicates that, despite the complex orography of the regions under study, precipitation evolves very coherently across Switzerland. Still, there are noticeable exceptions that appears as bands with more greenish and reddish colours. In Winter, region 4 in the observations, 3 in WRF-ERA and 2 in WRF-CESM exhibit the lowest correlations, reaching 0.2 in certain combinations of regions. Comparing with Fig. 2, these regions are located south of the Alps, and largely correspond to southern Switzerland, which stand out as regions with a remarkable, different behaviour. Similarly, in Spring the regions most strongly detached to the behaviour of the rest are 4 and 5 in the observations, 4 and 5 in WRF-ERA and 2 and 5 in WRF-CESM, which again correspond to the same Southeastern part of the country (see Fig. 2). In Summer, the Northwest-Southeast separation is still apparent and similar in both simulations (region 5 in both simulations, which corresponds to Ticino, is the most clearly decoupled), while such differentiation, although qualitatively similar, is not so strong in the observations, which exhibits correlations of up to 0.6 with region 1 in the Northeast. Finally, in Autumn the number of regions in both simulations is different (6 and 4) in WRF-ERA and WRF-CESM, respectively. However, the correlations in the bottom row in Fig. 3 show that this apparently different regionalisation can be understood in the same terms of Northwest-Southeast separation, as regions 4, 5 and 6 in WRF-ERA are the counterpart of region 2 in WRF-CESM, and the three formers behave collectively as the latter in terms of separation with respect the rest of the domain. The observations also reproduce this pattern in Autumn, although less clear, as correlations between regions are never below 0.4.

The skill of the WRF-ERA regarding its ability to reproduce the temporal evolution of observed precipitation in the period 1979-2005 is explored through a Taylor diagram that compares this dataset to the observations. Note that in this case the comparison with WRF-CESM is not meaningful due to the lack of assimilation of observations in the CESM simulation, therefore we skipped that dataset in the following analysis. The skill is assessed for each regional series, separately. This generates an inconsistency that complicates the calculation, as the number and shape of regions are different for the observations and WRF-ERA (see first and second columns in Fig. 2). We solve this by using the same regions to obtain the regional series in both datasets, which correspond to the ones obtained with WRF-ERA (second column in Fig. 2). The assessment of the skill is shown in Fig. 4, which depicts the results for each season (symbols) and region (colors). Daily correlations between WRF-ERA and OBS range between 0.6 and 0.9 in all cases, with an average of 0.78 (0.74 for Summer and 0.83 for Winter, respectively). This supports the lack of systematic errors attributable to driving conditions. Differences also appear in the ability of the simulation to mimic the temporal variability of precipitation. Region 1, which represents fairly consistently the central plains of Switzerland in all seasons, is where the agreement between the simulation and observations is best, with a ratio of standard deviations close to one. In the rest of regions, the model overestimates the variance about 20% compared to the observations. Part of this bias can be explained in terms of the systematic overestimation of precipitation through the annual cycle in the WRF-ERA simulation described in the next section. However, a striking feature is the severe overestimation of simulated precipitation in region 4 in Winter, which corresponds to a cluster that is only identified in the simulation, and spans the highest mountains in the Alps (see Fig. 2). As argued above, the observations in such locations are generally less reliable and are more strongly affected by extrapolation artifacts (due to data sparsity), and therefore a plausible explanation for this outlier is the underestimation of actual precipitation and its variance in the observational product.

In summary, the regions identified in both simulations are similar and resemble the orographical barrier imposed by the Alps. This similarity demonstrates that the spatial structure of precipitation regimes are largely independent on the driving dataset. This spatial structure is similarly reproduced in the observations, although boundaries are more sharply defined and correlations among regions are slightly larger (see for example the lack of correlations below 0.4 in Summer, or 0.3 in Autumn). The more pronounced differentiation of regional characteristics in the simulations compared to the observations might be explained by the effectively coarser resolution of the observational gridded product of precipitation. Moreover, the Taylor diagram demonstrates the acceptable performance of the WRF-ERA simulation as a plausible surrogate of the evolution of precipitation in Switzerland during the ERA-Interim period.

## 4.2 Climatology and annual cycle

In this section we compare the downscaled precipitation driven by ERA-Interim and CESM to observations to identify systematic model deficiencies leading to biases of the downscaled precipitation (Figs. 5 and 6). Figure 5 shows the precipitation averaged over Switzerland separately for each month, thereby emphasising the annual cycle, whereas the Figure 6 presents the maps of accumulated precipitation for each season (by columns) and dataset (columns 1 to 3).

The seasonality of precipitation is well reproduced by the WRF-ERA simulation (see blue bars in Fig. 5, as well as first and second columns in Fig. 6), showing a peak in the Summer months June to August and the driest months in Winter.

However, the WRF-ERA simulation generally overestimates precipitation throughout the year, in particular during December and January, which can be linked to the overestimation of precipitation variability identified in the previous section. This overestimation is especially noticeable in the highest locations around the Alps, but given the, in principle, larger uncertainties in the observations of precipitation in these locations, it is hard to judge to what extent this difference is directly attributable to just model deficiencies. In this regard, it is worth to note that there is a high agreement between WRF-ERA and OBS at low altitudes and valleys. Despite the general wet bias, the model underestimates precipitation in Ticino in Autumn. Isotta et al. (2014) show that in the region of Ticino up to 70% of the yearly precipitation accumulation is due to the top 25% of the wet days, so it is sensible to assume that the bias stems from high to extreme precipitation events. In Ticino these heavy precipitation events are driven by the transport of moist and potentially unstable (moist neutral stratification) air masses against the Alps from the south (Martius et al., 2006; Froidevaux and Martius, 2016). Locally, the vertical shear between south-easterly flow near the surface and southerly to southwesterly above 850 hPa leads to moisture convergence and repeated formation convective cells (Panziera et al., 2015). On an even more local scale, strong vertical shear can result in small-scale circulation that results in local precipitation maxima (Houze et al., 2001). Therefore if the RCM fails to capture any of these local and highly driven by the orography processes properly, it will result in an underestimation of the precipitation. The simulation is able to capture great part of the complex spatial structure of the climatology of precipitation which is induced by the complex topography (Fig. 6). The spatial correlation between the simulated and observed patterns (Fig. 6) lies between 0.78 (in Winter) and 0.84 (in Summer). These results can be compared to those obtained with an ensemble of RCM simulations driven by ERA-Interim within the EURO-CORDEX and MED-CORDEX projects. Fig. 2 in Fantini et al. (2016) is similar to Fig. 6 here, although the model resolution and observational gridded product used to validate the models are different. Further, Fig. 5 in Fantini et al. (2016) shows similar annual cycle as Fig. 5 here, but the Alps domain they consider is considerably larger, including western France, great part of Austria and the northern half of Italy. The comparison of these figures shows strong agreements, e.g. the simulations reproduce an orographical pattern with the highest precipitation over the Alps, they consistently overestimates precipitation, and they closely follow the annual cycle with the respective observational product. However a remarkable difference is that the annual cycle in the Alps domain in Fantini et al. (2016) presents a bi-modal curve without the unique and clear summer maximum we find for Switzerland and is consistent between WRF-ERA and the observations. Since the observational products are both of high quality and similar characteristics, this discrepancy is attributable to the disparity between the domains both studies consider.

As expected, the performance of the simulation when WRF is driven by CESM is lower (see red bars in Fig. 5, and first and third columns in Fig. 6). WRF-CESM shows strong deviations in the seasonal cycle with a maximum of precipitation in the extended Winter season from November to March and a strong underestimation of precipitation in Summer (Fig. 5). Strikingly, this behaviour is reverse to the observations, which show a peak in the Summer months from June to August and less precipitation in Winter. The spatial disaggregation of these biases are further explored in the seasonal precipitation patterns in Fig. 6. WRF-CESM strongly overestimates precipitation at high altitudes in Winter beyond the problems already stated regarding WRF-ERA. Further, it severely underestimates Summer precipitation (spatial average of 429.94 mm in the observations vs. 195.76 in WRF-CESM, respectively), without a clear footprint of orography in this bias. The spatial correlations between the

simulated (WRF-CESM) and observed patterns, although lower than in WRF-ERA, are still fairly high, ranging from 0.55 (in Autumn) to 0.78 (in Summer). Again, this correlation is due to the strong influence of orography. This further emphasises how the spatial distribution of precipitation regimes are, to a great extent, imposed by the RCM setup alone, whereas the ability of the simulation to reproduce the annual cycle is largely governed by the driving conditions provided externally through the boundaries. The performance of WRF-CESM can be compared to ESM-driven simulations within the EURO-CORDEX and MED-CORDEX ensembles. Figs. 3 and 4 in Torma et al. (2015) show the averaged winter and summer precipitation in the observations and the ensemble mean, and provide results consistent with the discussion about the influence of orography on precipitation presented above. Fig. 2 in Torma et al. (2015) shows the annual cycle for the same Alps domain employed by Fantini et al. (2016). The ensemble mean of ESM-driven simulations does reproduce the bi-modal annual cycle present in the observations for this domain, and the overestimation of precipitation is similar to the one obtained with these models are driven by ERA-Interim (Fantini et al., 2016). Therefore the seasonality biases of WRF-CESM seem not to be a general problem across ESM-driven simulations, but rather an issue specific to this ESM.

An important outcome of these simulations is the potential application to the study of extreme events. This type of study demands the disaggregation of precipitation into shorter periods than monthly averages. Although the daily correlation between WRF-ERA and OBS was shown in the Taylor diagram in Fig. 4, the ability of WRF to reproduce daily precipitation has not been explicitly analysed so far. Therefore, we evaluate model biases at daily scale by showing the Probability Density Function (PDF) of daily precipitation averaged over Switzerland for each season (Fig. 7). The overestimation of Winter precipitation in the WRF-CESM simulation stands out as an underestimation of the frequency of days with precipitation below 5 mm, i.e. the so-called "drizzling-effect", and its counterpart in the higher frequency of precipitation above 10 mm. WRF-ERA behaves similar to WRF-CESM, although the magnitude of this bias is lower. In Summer, the WRF-ERA simulation is able to mimic the distribution of precipitation. The WRF-CESM simulation exhibits a distorted PDF of daily precipitation in Summer, as the frequency of days with precipitation below 3 mm is strongly overestimated. This leads to the severe underestimation of precipitation apparent in Fig. 6. The comparison with the simulation driven by ERA-Interim, as well as the aforementioned results within the EURO-CORDEX ensemble (Fantini et al., 2016), show that this systematic error becomes attributable to biases in the boundary conditions provided by the CESM model. In the intermediate seasons of Spring and Autumn, both simulations exhibit an mixed behaviour, and their skill is remarkably good in Spring. Indeed, WRF-CESM allegedly outperforms WRF-ERA in Autumn. However the latter is not a demonstration of model performance, but an error cancellation artifact, as can be shown evaluating the performance through moving seasons (not shown). The behaviour of biases during this season are a combination of the ones in Summer and Winter, which are opposite and therefore tend to cancel out when pooled to obtain the PDF.

## 5   Bias correction of the WRF-CESM simulation

From the results described so far, three important conclusions can be drawn:

- WRF-ERA mimics many important features of the observed spatio-temporal distribution of precipitation, even at daily scale and through the annual cycle.

- The spatial structure of precipitation variability is strongly affected by orographic features, and is prescribed by the RCM. This leads to consistency between WRF-ERA and WRF-CESM, and together with the first point, supports the reliability of the latter simulation.

- The temporal evolution is driven by the boundary conditions, and in particular the WRF-CESM presents important systematic biases through the annual cycle that can not be removed with dynamical downscaling alone.

These conclusions together suggest that although the output of the WRF-CESM is a valuable resource with potential applications, it might be desirable to post-process this dataset in a way that systematic biases are ameliorated. Therefore, the new bias correction method binding cluster analysis and quantile mapping (Sec. 3) is applied to the WRF-CESM simulation.

The results of the bias correction method are presented in the Figs. 5 to 7 showing the desired improvements: the mean precipitation fields agree better with the observations, so that the annual cycle is corrected in a way that closely follows the observed values (green bars in Fig. 5). In particular, the strong overestimation (underestimation) in Winter (Summer) has been removed to a large extent. It is worth to note the clear improvements in the ability of the bias-corrected dataset to mimic the annual cycle compared to the results obtained with a simpler method that does not account for the spatial heterogeneity (Fig. 5 in Felder et al., 2018), as well as in the spatial patterns of precipitation (Fig. 4 in Felder et al., 2018). The bias correction also improves the intensity of precipitation and preserve its spatial structure (compare second and fourth columns in Fig. 6). This is important, as according to the results above, this structure is in agreement with the more reliable WRF-ERA simulation. However, it does not improve the spatial correlation with the observations, which ranges between 0.54 (in Autumn) and 0.78 (in Summer). Interestingly, an improvement is also found on a daily scale (green curve in Fig. 7). The underestimation of the frequency of days with very low precipitation in Winter is corrected, although it leads to a slight overestimation. This effect occurs when models tend to underestimate the dry-day frequency, as all days become mapped onto a precipitation day, producing a wet bias. This could be further corrected using frequency adaptation techniques (Themeßl et al., 2012), although we have not considered such techniques here. Above 5 mm the precipitation PDF is remarkably well captured. Similarly, in Summer the bias correction improves the PDF, although does not completely remove the overestimation (underestimation) of the frequency of dry (wet) days; above 4 mm the simulated PDF is barely indistinguishable from the observed one. Again, intermediate seasons exhibit a mixed behavior. In Autumn, the PDF of bias-corrected WRF-CESM simulation is apparently worse than the uncorrected WRF-CESM simulation. This reinforces the argument developed above regarding the apparent skill of the simulation in this season due to error cancellation.

As the proposed bias correction employs a non-linear transformation on a daily basis, which is based on a transfer function that differs for each month within the annual cycle, it does not simply scale precipitation, but modifies it in a complex manner. Such modification slightly changes the temporal evolution of precipitation at every grid point. This is an undesired side effect, as the temporal co-evolution of all simulated variables is bounded by the equations being solved by the model, and therefore modifications to this evolution may underscore the most valuable aspects of the dynamical downscaling: its physical

consistency (Maraun, 2016). This effect is unavoidable, it depends on factors such as the magnitude of the biases, their location within the precipitation distribution, or their variability through the annual cycle, and should be ideally kept to a minimum. We demonstrate how the applied bias correction has only slightly affected the temporal evolution in Fig. 8, which shows the daily correlation separately by seasons to avoid the overestimation of correlation due to the annual cycle. The point-wise correlation between the raw and corrected simulation is well above 0.8 in all seasons across the domain, and lower than 0.9 in Autumn in just few quasi-random locations. The lower correlation in this season is motivated by the larger variability of the nature of biases within this season, which drives a large spread between the transfer functions for the three months, and therefore reduces the linear relationship between raw and corrected series (not shown). There is no obvious indication in these maps of geographical influences, e.g. orographic, longitudinal, etc. that might point out systematical errors attributable to a misrepresentation of physical processes at regional scales.

## 6 Conclusions

This study presents the performance and biases of two high-resolution climate simulations, and introduces a new bias correction technique that reduces systematic biases based on the regionalisation of precipitation. The simulations span the recent past 1979-2005 over the entire Alpine region, although we limit the analysis and bias correction of the simulation to the area of Switzerland due to the limited spatial coverage of the observational product we use as reference. Both simulations are carried out with a RCM driven by two global datasets, an ESM (CESM) and a reanalysis product (ERA-Interim). The bias correction is based on quantile mapping, but it is separately applied to different regions of common variability, which are identified by objective cluster analyses.

   The comparison between simulations and observations shows that regions of common variability agree between the two simulations and to a great extent with the observations. Still, the observed regions of common variability lack of many fine details found in the simulations due to the coarser effective resolution RhiresD data and potentially the sparse data network at high altitudes. Besides the regional classification, further agreements and differences between the simulations and observations are found. The WRF-ERA simulation is able to simulate the seasonal cycle but consistently overestimates precipitation by about 20%. The day-to-day variability is captured by the WRF-ERA simulation with rather high positive correlation, but the simulated variability is again larger than in the observations. At least for Winter, overestimation of simulated variance is related to a potential underestimation of observed precipitation due to the sparsity of observations in high mountains. The biases of the WRF-CESM simulation are expected to be larger as the driving CESM data do not incorporate observations. The WRF-CESM simulation is not able to simulate the seasonal cycle correctly with a strong overestimation (underestimation) of Winter (Summer) precipitation.

   To correct for these systematic biases a new bias correction technique is applied to the WRF-CESM simulation. The separation in regions of common variability by the cluster analysis acknowledges the fact that biases in different regions and seasons are produced by different physical mechanisms, and minimises the risk of error cancellation. This method clearly improves simpler approaches that do not account for this heterogeneity, and is an issue when quantile mapping is applied to larger regions

like the entire Switzerland (Felder et al., 2018). The spatial structure of bias corrected precipitation is preserved compared to the original WRF-CESM, but the seasonality is corrected in a way that nearly mimics the observations. This improvement is also found when analysing the daily scale. This means that the temporal evolution of the simulation, which emerges from the physical consistency of the simulation, is greatly preserved, as the daily temporal correlation between the raw and corrected

versions of the WRF-CESM simulation is above 0.9 in most cases, except for few quasi-random grid-points in Autumn.

We note that the rationale of the developed methodology is to divide a large domain into smaller subregions according to the behaviour of the target variable. We have applied it here to daily precipitation in Switzerland for being a variable strongly affected by complex orographical details that lead to strong horizontal gradients. With more generality, spatial regionalisation is an efficient method to break down complexity in areas and variables whose behaviour strongly varies through the domain.

Still, the bias correction applied separately to subregions can be in principle adapted to other cases with simpler topography, or other variables with lower horizontal gradients. The only practical difference is that in this case the regionalisation will naturally lead to a lower number of subregions which are necessary to obtains clusters with coherent features.

Finally, the applicability of the three datasets, i.e. WRF-ERA, the raw WRF-CESM, or the corrected version of WRF-CESM, depends on the nature of the question to be addressed. For applications where a match with the actual observed climate

is needed, the ERA-Interim driven simulations is suitable. However, there are research questions for which a simulation driven by an ESM, such as WRF-CESM, is necessary. This is for example the case for climate change projections, but also climate simulations of past conditions, or studies of extreme situations in long simulations (Felder et al., 2018) or sensitivity studies (Messmer et al., 2015). Finally, the use of corrected variables is advisable only when an accurate simulation of the magnitude of the variable under consideration is critical for the application. An example is the use of output of climate simulation as input

in hydrological modelling (Camici S. et al., 2014; Felder et al., 2018), as the magnitude of rainfall in a given location, and not only its large-scale structure or temporal consistency, is crucial for an realistic simulation of river discharge.

*Code availability.*  All code used through this manuscript is open source. WRF is a community model that can be downloaded from its webpage (http://www2.mmm.ucar.edu/wrf/users). The code to perform the regionalisation, as well as the Taylor diagram, is based on R and Bash scripts, whereas quantile mapping and PDF estimation is implemented with Fortran 90. The source code of these tools is available in a Github

repository (https://github.com/Onturenio/BiasCor). Simple calculations carried out at each grid point, e.g. means, correlations, etc. have been performed with CDO (https://code.mpimet.mpg.de/projects/cdo). The figures have been prepared with GMT (http://gmt.soest.hawaii.edu)

*Data availability.*  The CESM simulation was carried out at the University of Bern, and is available once approved by the original authors. The ERA-Interim dataset can be downloaded from the ECMWF webpage, although it requires previous registration. The two datasets produced, WRF-ERA and WRF-CESM consists of hourly output of a number of variables, and therefore occupies several Terabytes and is not freely

accessible. Still, it can be accessed upon request to the authors of this manuscript.

*Author contributions.* JJGN coordinated the work, carried out the WRF-ERA simulation and the calculations of this manuscript. CR contributed in the design of the simulations and their analysis. DB carried out the WRF-CESM simulation. OM helped in the design of the simulations and the discussion of the results. JAGV provided the code to carry out the regionalisation and helped in its analysis. JPMG provided ideas for new approaches in the analysis of the simulations that have been integrated in the final manuscript. The manuscript has been written by JJGN and CR, and all authors have contributed reviewing the text.

*Competing interests.* The authors declare that they have no conflict of interest.

*Acknowledgements.* This work is supported by the Mobiliar Lab and the Swiss National Science Foundation (grant pleistoCEP (Nr. 200020_159563)). The CESM and WRF simulations were performed on the super computing architecture of the Swiss National Supercomputing Centre (CSCS). JJGN acknowledges the CARM for the funding provided through the Seneca Foundation (project 20022/SF/16). Special thanks are due to Flavio Lehner who performed the seamless transient simulation with CESM.

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

**Table 1.** Number of regions obtained after the cluster analysis of daily precipitation. The shape of such regions is shown in Fig. 2. The number of EOFs retained is kept to 7 in all cases, which corresponds to a explained variance above 80% in all cases.

|       | OBS | WRF-ERA | WRF-CESM |
|-------|-----|---------|----------|
| DJF   | 5   | 4       | 6        |
| MAM   | 5   | 5       | 6        |
| JJA   | 5   | 5       | 5        |
| SON   | 5   | 6       | 4        |

**Table 2.** Number of grid points that belong to each of the regions shown in Fig. 2. Only grid points within the Swiss domain, i.e. those not missing values in OBS, are counted. Note that in some cases the number of regions is lower than 6, therefore we indicate it with ah dash.

|         | DJF  |      |      | MAM  |      |      | JJA  |      |      | SON  |      |      |
|---------|------|------|------|------|------|------|------|------|------|------|------|------|
|         | OBS  | WE   | WC   | OBS  | WE   | WC   | OBS  | WE   | WC   | OBS  | WE   | WC   |
| Reg. 1  | 1233 | 1830 | 1719 | 1017 | 1956 | 1800 | 897  | 1746 | 1293 | 1116 | 1812 | 2193 |
| Reg. 2  | 1203 | 954  | 579  | 837  | 471  | 438  | 846  | 579  | 618  | 945  | 492  | 693  |
| Reg. 3  | 738  | 564  | 372  | 825  | 708  | 678  | 822  | 777  | 786  | 786  | 747  | 606  |
| Reg. 4  | 375  | 435  | 708  | 771  | 426  | 294  | 735  | 327  | 630  | 471  | 291  | 291  |
| Reg. 5  | 234  | –    | 345  | 333  | 222  | 246  | 483  | 354  | 456  | 465  | 183  | –    |
| Reg. 6  | –    | –    | 60   | –    | –    | 327  | –    | –    | –    | –    | 258  | –    |

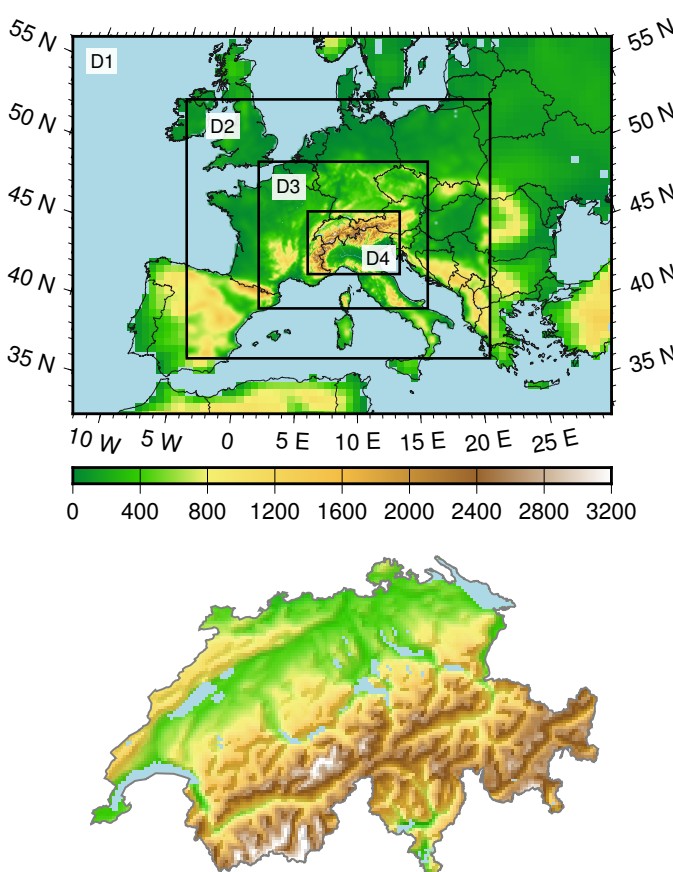

**Figure 1.** Top: configuration of the four nested domains used in both the WRF-ERA and WRF-CESM simulations. Bottom: detail of the actual orography implemented in the 2-km resolution simulation over Switzerland.

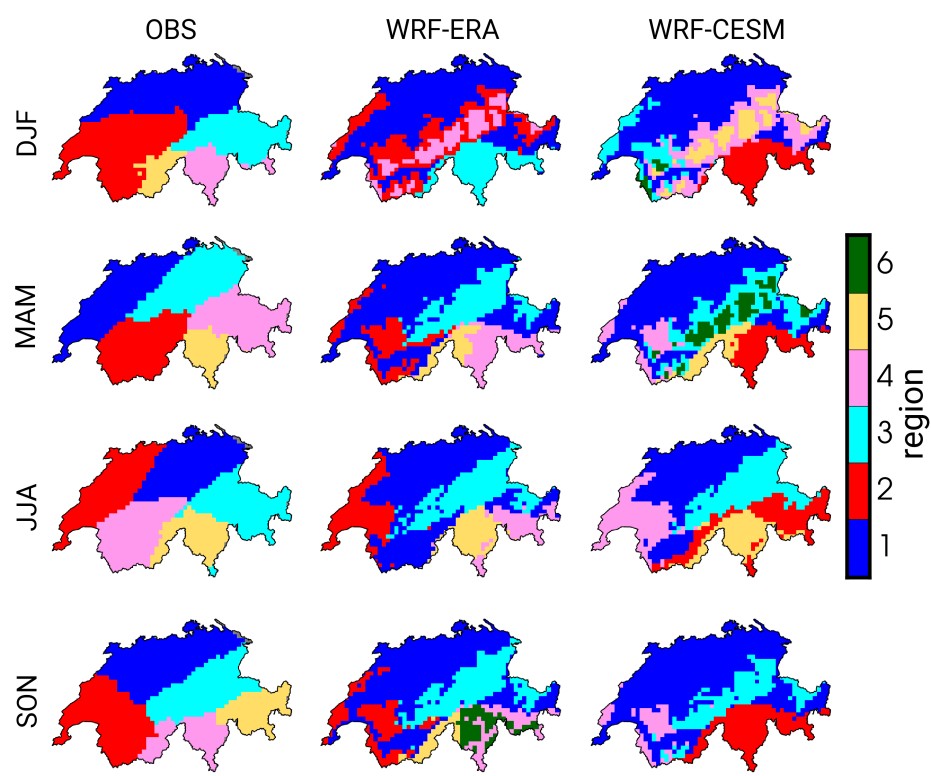

**Figure 2.** Regions obtained from the cluster analysis described in Sec. 3. The maps correspond to the 12 possible combinations, 3 for each dataset (OBS, WRF-ERA and WRF-CESM) and 4 for each season. Note that the colors are set arbitrarily as a label within the algorithm, so no one-to-one correspondence is implied between regions of the same colour in different maps.

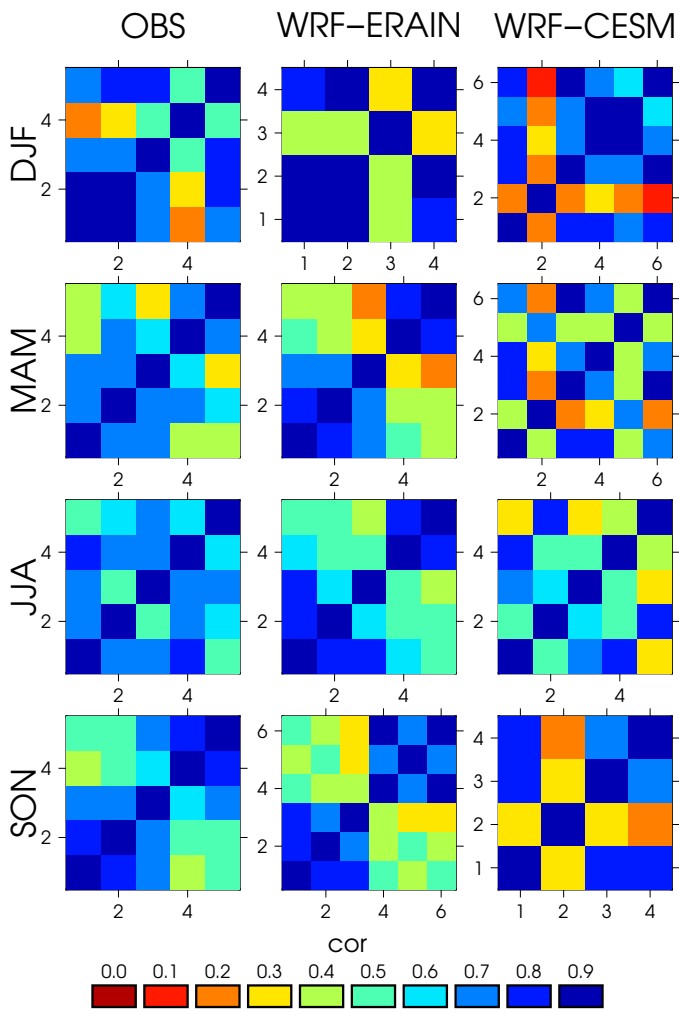

**Figure 3.** Temporal cross-correlation matrices between all regional series. The calculation, as the definition of regions, is carried out independently for each dataset and season. The order of matrices is from region 1 (bottom-left) to region 6 (top-right), and the spatial distribution of the regions is shown in Fig. 2. Note that all matrices are symmetric with 1 across the diagonal.

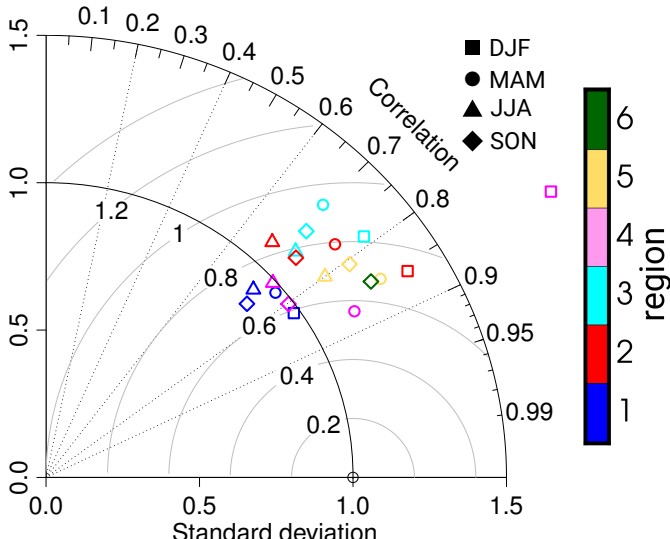

**Figure 4.** Taylor diagram showing the temporal correlation and ratio of standard deviation between the regional series in the WRF-ERA simulation and the observations across all 4 seasons. For obtaining the regional series, the regions defined for WRF-ERA are used in both datasets. Different symbols denote the result for each season, whereas the colours correspond to the different regions according to the legend and spatial structure shown in middle column in Fig. 2.

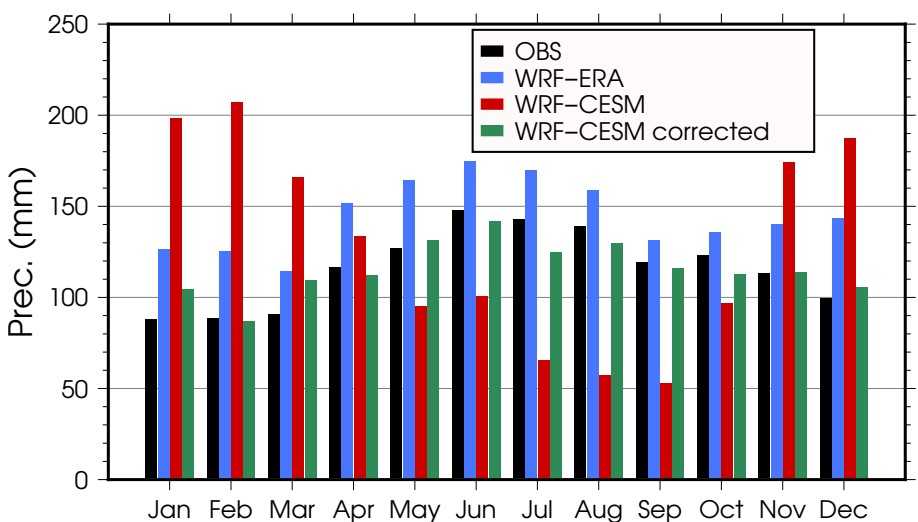

**Figure 5.** Seasonal cycle of monthly precipitation over Switzerland in the observations (black), the WRF-ERA simulation (blue), the WRF-CESM simulation (red), and bias-corrected WRF-CESM simulation (green).

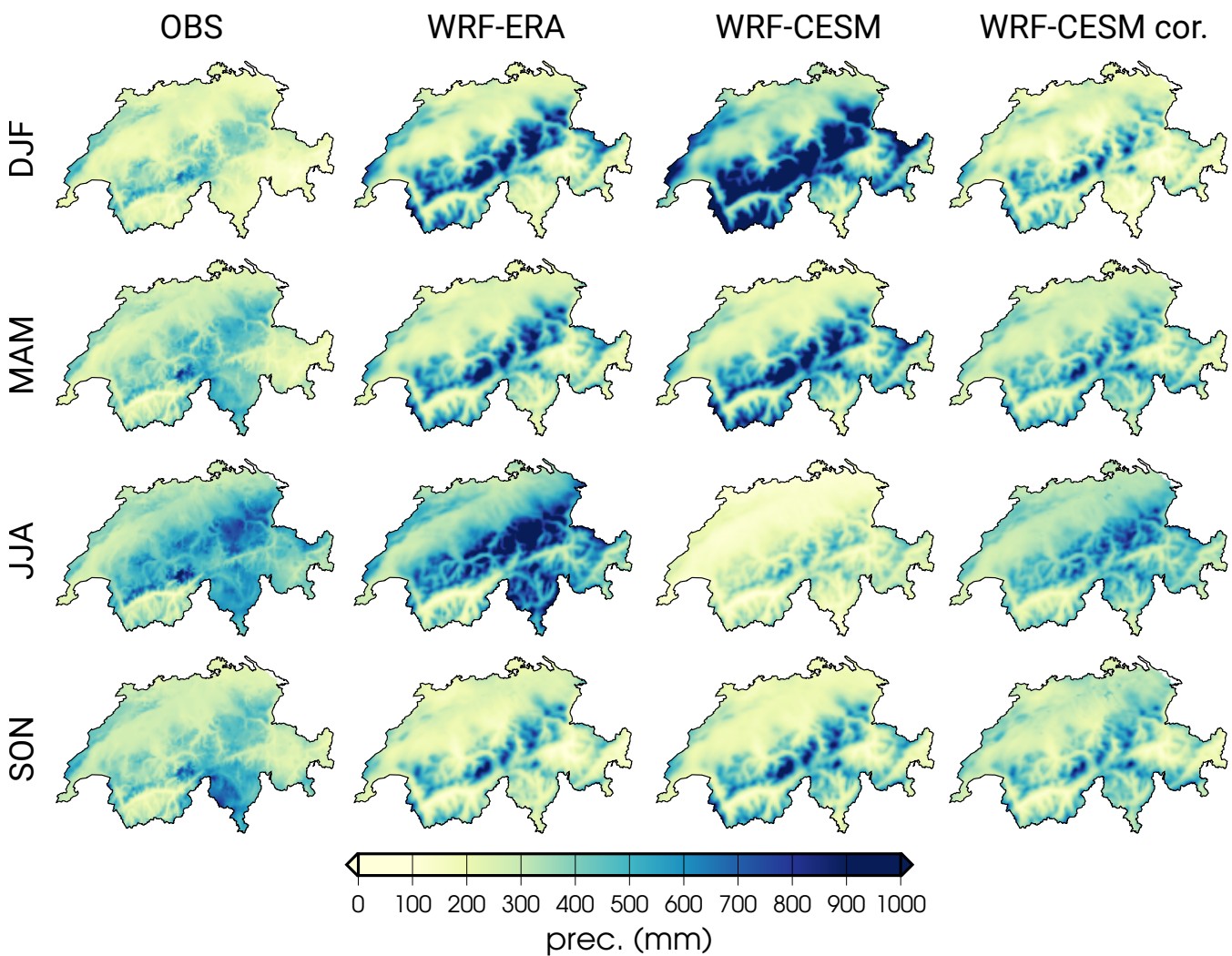

**Figure 6.** Mean seasonal accumulated precipitation over Switzerland across seasons (different rows) in the gridded observations (first column), in the WRF-ERA simulation (second column), in the WRF-CESM simulation (third column) and the bias-corrected WRF-CESM simulation (forth column).

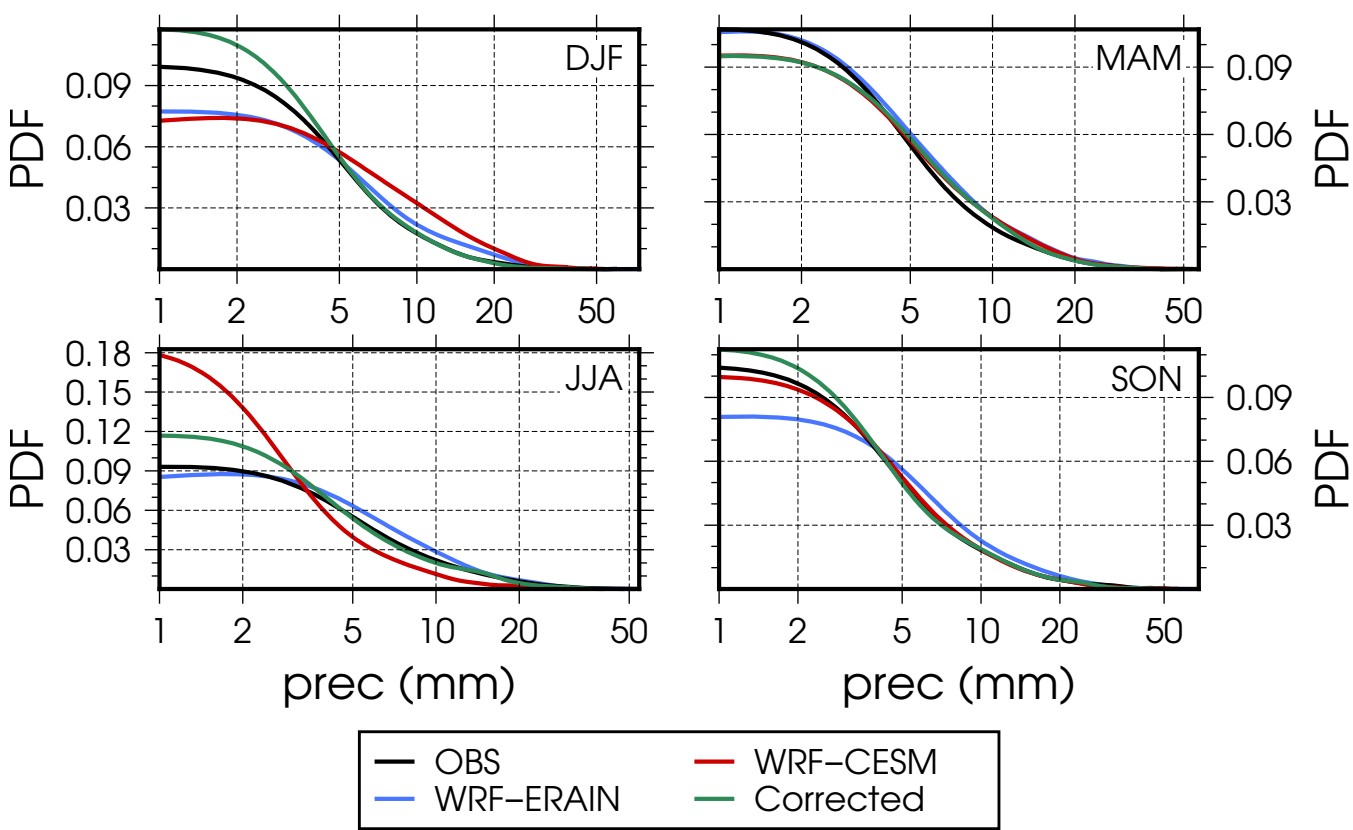

**Figure 7.** Estimated PDFs of daily precipitation averaged over Switzerland. Each panel depicts the result for a season, and different colors are representative of the results for different datasets according to the choice in Fig. 5. Note the logarithmic scale in the x axis, which precludes the area below all curves being equal.

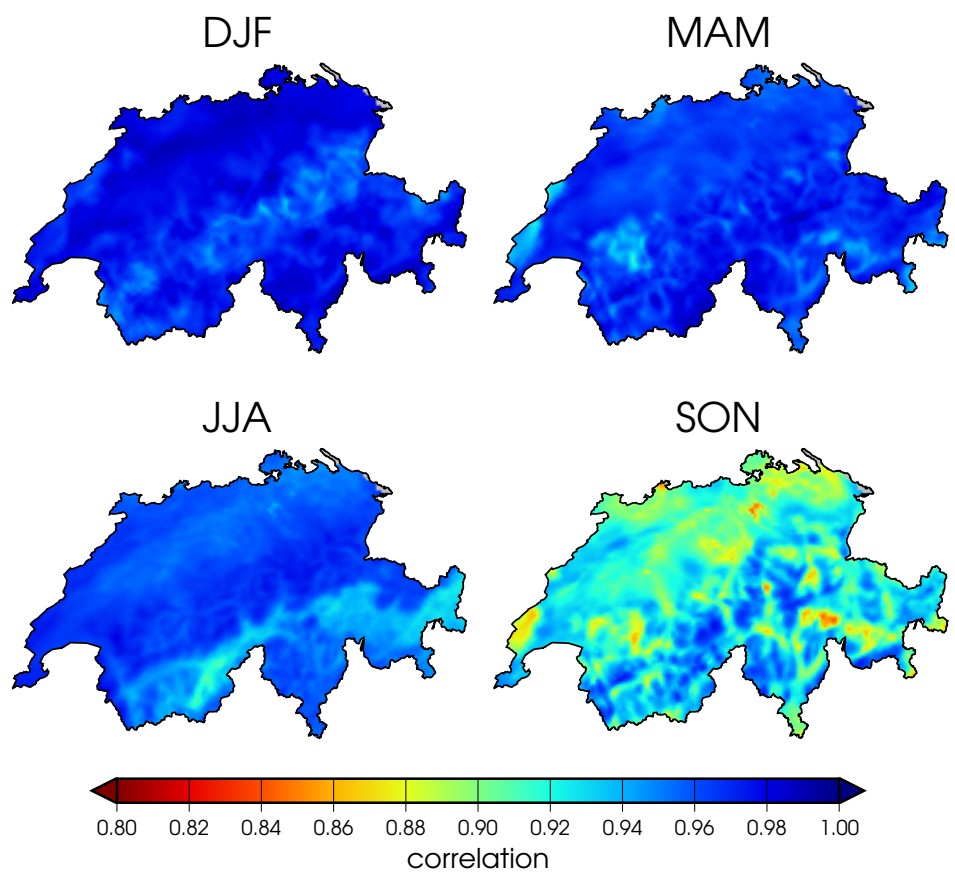

**Figure 8.** Correlation maps between the daily series of precipitation in the raw WRF-CESM simulation and the output of the bias corrected. The analysis is carried out separately by seasons to minimize the effect of the annual cycle on correlation.