# Peer review of "A new region-aware bias correction method for simulated precipitation in areas of complex orography"

_Geoscientific Model Development, 2017_

## Referee Comment (RC1) · Anonymous Referee #1 · 4 Apr 2018

General comments:

The paper is well written and addresses a relevant scientific question by describing a promising bias correction method, based on quantile mapping (QM) conditioned to regions with similar temporal variability. It is in general well-structured and represents a substantial contribution to the modelling and impacts community. Still there are some explanations missing to be able to understand the whole methodology and these explanations may probably answer some of my specific comments. In particular, the regions/clusters are obtained for observations and model independently, I do not understand how the bias correction is trained and applied for each grid box, since the

regions are different for each dataset and a grid box may belong to different clusters in both datasets. Thus, how are the calibrated corrections obtained for a region? Which correction is applied to a grid box that belong to different regions in the model and observations? And some further issues:

1) If the cluster classification of the raw model data is used, this classification is based on biased data.

2) If only the classification for the train/test of the QM based on observations is used, how would be the method applied in a changing climate in which the grid boxes could move to another cluster?

3) Can one relate those "objective" clusters to e.g. hydrological catchments relevant for impact studies?

4) How does the different number of grid boxes in each cluster affect the results? The authors may include the number of grid boxes per region in Fig.2.

A further concern is if the authors checked differences/improvements with respect to standard QM (without conditioning to regions). Some discussion about this would be appreciated.

Here I list some specific comments and typos, giving the page and line numbers.

Specific comments:

P1 L6 "minimise disturbances to the physical consistency" -> not clear, please rephrase or elaborate.

P1 L16 which variables? So far only precipitation was mentioned (also in the title). If the clustering depends on the variable, why does the method preserve the physical consistency among variables more than the standard QM?

P2 L3 The authors may consider citing the newer analysis including EURO-CORDEX data: Rajczak, J. and C. Schar (2017), Projections of future precipitation extremes over

Europe: a multi-model assessment of climate simulations | J. Geophys. Res. Atmos., doi:10.1002/2017JD027176.

P3 In the review of bias correction methods, the authors may consider the following paper, with some similarities from a technical point of view, where the bias correction is conditioned to circulation types: Wetterhall, F., Pappenberger, F., He, Y., Freer, J., and Cloke, H. L.: Conditioning model output statistics of regional climate model precipitation on circulation patterns, Nonlin. Processes Geophys., 19, 623-633, https://doi.org/10.5194/npg-19-623-2012, 2012.

P3 L15 After this paragraph I suggest to include a sentence mentioning the implications in the climate change context, something like "As a consequence, the climate change signal might be unrealistically modified", as stated e.g. by:

Casanueva, A., Bedia, J., Herrera, S., Fernández J. and Gutiérrez J.M. Direct and component-wise bias correction of multi-variate climate indices: the percentile adjustment function diagnostic tool. Climatic Change (2018) 147: 411. https://doi.org/10.1007/s10584-018-2167-5

Teng J, Potter NJ, Chiew FHS, Zhang L, Wang B, Vaze J, Evans JP (2015) How does bias correction of regional climate model precipitation affect modelled runoff? Hydrol Earth Syst Sci 19(2):711–728. https://doi.org/10.5194/hess-19-711-2015

P3 L16-20 The authors may consider the above paper (Rajczak and Schar 2017) to update that summary of previous works.

P3 L20-21 what do the authors mean with "similar"? different model version? Parameterizations?

P3 L16-25 I would suggest to move the entire paragraph before the previous one, in which bias correction is introduced, since it reads better after line 3 and here it is again about previous studies in which bias correction is not applied. Also the final lines of the paragraph (23-25) are more or less repeating what it is already said in P2 L34.

P5 L24 Is there a reason for using 27 years instead of e.g. 30?

P7 L7 I suggest to add "smooths out the transfer functions prior to the correction".

P7 L9 Until now it is not clear which is the analysis domain, the title says Alpine region, simulations are performed for the Alpine region but observations are available for Switzerland. Please consider to mention Switzerland explicitly in the experimental design and title.

P7 L16 As mentioned before, the authors mention the preservation of physical consistency. My question now is how coherent is the method in a multivariate case? I guess a different division in clusters would be performed for each variable. Can the authors comment something on this?

P7 L21 "varies per season" why seasonally? In section 3 it is said that the method is applied for each month, thus one expect to have different clusters at the monthly scale.

P8 L13 The authors mention several times the insufficient effective resolution of the observations, what about the effective resolution of the simulations? The authors should include it in the discussion as well.

P8 L15-35 The authors should motivate better the correlation analysis in Fig.3. I do not see the point of this analysis, especially since the clusters are built in a way that the differences among clusters are maximized. Moreover, the clusters are different in each dataset, so there is not a clear correspondence. This lack of correspondence is only mentioned and resolved in Fig.4.

P9 L31 "averaged over Switzerland" Given the differences in the annual cycle among the regions, the authors may consider doing this analysis per cluster, based on the observations or the WRF-ERA classifications.

P10 L1-8 The underestimation of precipitation in the Ticcino during autumn is worth to mention. Can the authors give a reason for this?

P10 L12 The authors should also explicitly mention in the methods how the precipitation frequency is adjusted by this method (relevant for the interpretation of Fig.7). Standard QM is able to correct for a higher frequency of wet days in the model, but the opposite problem (here shown in Fig. 7, winter) could be corrected by applying the frequency adaptation, otherwise an overestimation of the wet day frequency is found in the corrected data. See : Themeßl, M.J., Gobiet, A. & Heinrich, G. Empirical-statistical downscaling and error correction of regional climate models and its impact on the climate change signal. Climatic Change (2012) 112: 449. https://doi.org/10.1007/s10584-011-0224-4

P12 L4 Why are the temporal correlations lower in autumn? This may be related to the way the corrections are trained and applied.

P22 Fig.4 The decimal dots are missing in the labels of the Taylor diagram. And more important than that, it is completely unclear to me what is shown by the angular scale (azimuthal angle). I would expect to have represented there correlation values but that legend must be something else. Please explain in the caption how this should be interpreted.

Technical corrections:

P1 L1 Do the authors mean better than the coarse global models? I suggest to include better than what.

P1 L13 I think the sentence will read better with "Conversely, WRF-CESM shows a different seasonality..." This is only a suggestion.

P2 L2 I think it would be "burden with".

P2 L12 "precludes the simulation following ... " -> "precludes the simulation from following...".

P2 L25 "and in variables where the..." -> "and variables for which the ..."

P2 L28 "with the help of" -> not sure if "the" should be omitted.

P2 L32 Add "change in spatial resolution", since it is the first time that this issue is mentioned.

P3 L2 "a major source of uncertainties" -> not clear to which part of the previous sentence this is referring to. I suggest to add "being a major source of uncertainties..."

P4 L5 I think "the" in "bias-correct the precipitation" should be removed.

P5 L7 "analyses ... is" -> "analyses ... are".

P6 L17 "distances ... needs" -> "distances ... need".

P6 L32 "quantiles from 1st to 99th are corrected". Note that percentiles and quantiles are not the same, I think here the authors refer to 1st to 99th percentiles.

P9 L10 "differences also appears" -> "differences also appear".

P9 L12 Not sure if the authors should include "the" in "observations is the best".

P9 L22 Sentence "The spatial structure ..." Please rephrase this sentence. If the observations are considered as the truth, I do not understand that "the spatial structure is well reproduced" by them. "correlations are slightly larger", than what?.

P9 L25 "gridded product", the simulations are also gridded, so here I would add "observational gridded product".

P10 L14 "binning" -> do the authors mean "binding"?

P12 L10 "Alpine area" -> I would suggest to talk about Switzerland.

P13 L3 "questions where" I would say "questions for which".

Table 1 (caption) "variance explained" -> "explained variance".

---

## Referee Comment (RC2) · Anonymous Referee #2 · 7 Apr 2018

**Review for Geoscientific Model Development (Manuscript gmd-2017-329)**

**Title:** "A new region-aware bias correction method for simulated precipitation in the Alpine region"

**Authors:** J. J. Gómez-Navarro, C. C. Raible, D. Bozhinova, O. Martius, J. A. García-Valero, J. P. Montávez

[Figure]

**1 General comments**

This paper presents a bias correction method for regional climate simulations over the Alps at very high resolution. A observational database for the region is used for the validation, and ERAinterim and GCM-CESM forcing fields are used to WRF modelling work. To my opinion, it shows enough aspects to novelty and adequate analysis and understanding of the obtained results. I suggest it to be considered for publication, once the questions and requested item can be properly answered or at least taken into account in some way.

1. Missing references. It is always the case that not all the relevant references are included when a work is presented. Here I find some that I consider that are essential to be included, not only for the introductory aspects, but also for the methods and results description. Let me indicate them to the authors for them to be considered a properly used throughout the text

   (a) Torma, C., Giorgi, F., Coppola, E. (2015). Added value of regional climate modeling over areas characterized by complex terrain-Precipitation over the Alps. Journal of Geophysical Research: Atmospheres, 120(9), 3957-3972. This work should be mentioned because of similar modelling domain and resolutions are used, and for sure some of the figures there could be related to the results shown here.

   (b) Fantini, A., Raffaele, F., Torma, C., Bacer, S., Coppola, E., Giorgi, F., ... & Verdecchia, M. (2016). Assessment of multiple daily precipitation statistics in ERA-Interim driven Med-CORDEX and EURO-CORDEX experiments against high resolution observations. Climate Dynamics, 1-24. Here an ensemble of RCMs is used for the whole Europe, but some specific analysis over the Alps is seen.
(c) Giorgi, F., Torma, C., Coppola, E., Ban, N., Schär, C., & Somot, S. (2016). Enhanced summer convective rainfall at Alpine high elevations in response to climate warming. Nature Geoscience, 9(8), 584. Perhaps this specific work could also be included

Other works propose some clustering methods based on precipitation, or other bias correction procedures for precipitation fields as obtained from regional climate models, although perhaps not only for the alpine region, but that they maybe should be considered to be mentioned on this work:

(a) Casanueva, A., Kotlarski, S., Herrera, S., Fernández, J., Gutiérrez, J. M., Boberg, F., ... & Keuler, K. (2016). Daily precipitation statistics in a EURO-CORDEX RCM ensemble: added value of raw and bias-corrected high-resolution simulations. Climate dynamics, 47(3-4), 719-737.

(b) Dosio, A. (2016). Projections of climate change indices of temperature and precipitation from an ensemble of bias-adjusted high-resolution EURO-CORDEX regional climate models. Journal of Geophysical Research: Atmospheres, 121(10), 5488-5511.

(c) Argüeso, D., Evans, J. P., & Fita, L. (2013). Precipitation bias correction of very high resolution regional climate models. Hydrology and Earth System Sciences, 17(11), 4379.

(d) Argüeso, D., Hidalgo-Muñoz, J. M., Gámiz-Fortis, S. R., Esteban-Parra, M. J., & Castro-Díez, Y. (2012). High-resolution projections of mean and extreme precipitation over Spain using the WRF model (2070–2099 versus 1970–1999). Journal of Geophysical Research: Atmospheres, 117(D12).

(e) Manzanas, R., Lucero, A., Weisheimer, A., & Gutiérrez, J. M. (2018). Can bias correction and statistical downscaling methods improve the skill of seasonal precipitation forecasts?. Climate Dynamics, 50(3-4), 1161-1176.

2. Apart from the pure bibliography missing items, there are some aspects that could be more deeply described by the authors. One of them should be to compare the proposed bias correction method with other similar ones, if there are some, to see more clearly differences and similarities with others already proposed. I am sure the quantile mapping procedures have been used before, if one goes to those references. Therefore, I recommend the ongoing work by Nikulin and others in the frame of EuroCORDEX activities, named BCIP. Take a look at this abstract at EGU2015: Nikulin, G., Bosshard, T., Yang, W., Bärring, L., Wilcke, R., Vrac, M., ... & Fernández, J. (2015, April). Bias Correction Intercomparison Project (BCIP): an introduction and the first results. In EGU General Assembly Conference Abstracts (Vol. 17). In a more general sense, perhaps a mention to this recommendation by CORDEX community could be made. take a look at http://cordex.org/data-access/bias-adjusted-rcm-data/, and from there, to a IPCC work focused on this topic: See Breakout Group 3bis: Bias Correction (pp. 21-23) in IPCC, 2015: Workshop Report of the Intergovernmental Panel on Climate Change Workshop on Regional Climate Projections and their Use in Impacts and Risk Analysis Studies [Stocker, T.F., D. Qin, G.-K. Plattner, and M. Tignor (eds.)]. IPCC Working Group I Technical Support Unit, University of Bern, Bern, Switzerland, pp. 171. (https://www.ipcc.ch/pdf/supporting-material/RPW_WorkshopReport.pdf). I can imagine that authors do not want to go too far on this aspect, but I think that some more comments, to have this work inside the wider context, should be made. Even a mention to some developed software for this kind of analysis could be included, such as Bedia, J., Iturbide, M., Herrera, S., Manzanas, R., & Gutiérrez, J. (2017). downscaleR: an R Package for Bias Correction and Statistical Downscaling. R Package Version 2.0-3.

3. I am not sure if the authors have a comment about the fact that this bias correction method has been applied to a region with a very deep orography, and to precipitation field. Which could be the potential to apply it to other regions with

smoother orography, and/or to other variables?

**2   Specific comments**

1. It has been indirectly mentioned on the general comments section, but here I want to comment if explicitly: I miss a mention to the EuroCORDEX/MedCORDEX activities, that have used plenty of simulations at high resolutions (0.11) over Europe, and several studies with not a single RCM as here, but an ensemble of them, that have analyzed, also forced with ERAinterim fields, how precipitation is described. I do not mean a full comparison with other RCMs, but at least some mention and comparison with them, to see more clearly if WRF-RCM is similar to the state-of-the-art RCMs modelling alpine precipitation for current climate conditions.

2. And also related to this point, I miss some comparison of your figure 5, for example, with figure 2 of Torma et al., 2015 or Fantini et al., 2016, figure 5, not only for RCMs, but also for observational datasets, I am not sure if they are totally consistent. Or for your figure 6 and 7, and their corresponding figures.

3. I have a concern about the domain of study chosen here. On figure 1, D4 subdomain seems to be the one used for the analysis, but then figures with the political borders of Switzerland seem to be used. This relatively artificial borders could add some non-physical or modelling aspects to the analysis, and specially when obtaining the subregions from the clustering procedure. Which is the opinion of the authors about this aspect?.

4. Another point I would like to hear from the authors is about the very high resolution used for the WRF D4 domain (page 5, line 14): 2km. Which one is the real advantage here of using such resolution compared with the even-very-high

6km one?. It seems that no much mention or usefulness is made by the authors to this resolution, by far much larger than the mentioned 0.11 "high resolution" EuroCORDEX standard values these days. It is also a tricky aspect, since the comparison and bias correction method is made against the roughly 20km observational dataset information, and so some statements are made through the text related to this resolution differences. A more complete study should perhaps include at least some other resolution from the WRF model to a better understanding of the resolution topic?.

5. I understand that the forcing GCM is always an open question, but the usage of just one instead of, at least, a couple of them, does not limit a little bit the representativity of the GCM-forced RCM analysis?

6. The result shown in pages 10-11 that related intermediate seasons with cancellation artifacts sounds reasonable, but perhaps a more specific analysis could be made, with moving seasons, to see if more clear picture of that can be obtained. Because on the other hand, this result could be found non-intuitive, as one can think that precisely those transition seasons are more difficult to be properly captured. Which are the thoughts of the authors about it?.

7. Page 11, line 22. The bias corrected result over the frequency distribution that changes from underestimation to overestimation in winter looks a little bit peculiar. Could this result be a little bit further explained?

**3  Technical corrections**

1. When describing the experimental design (page 5, line 25) I do not understand those 6-day chunks and 12h spinup periods. I thought that a whole year or even two or more where needed for the soil moisture to be adapted. Could this aspect

be explained a little bit more? I understand that more details can be found in Gomez-Navarro et al., 2015, but perhaps here it is too little what is said. It is the same about D1-D2-D3-D4 subdomains and nesting aspects.

2. Close to this point, I do not also understand why nudging is applied to ERAinterim forced simulation, but not to the ESM one.

---

## Author Comment (AC1) · 15 May 2018

Thank you very much for the detailed review and very constructive comments. We have reviewed them and we think we can address them in a new version we are currently preparing. I attach to this comment a PDF document we have prepared with a point-by-point response to all the queries.

Please also note the supplement to this comment:
https://www.geosci-model-dev-discuss.net/gmd-2017-329/gmd-2017-329-AC1-supplement.pdf

**Supplement:**

**Point-by-point response to Reviewer #1**

J. J. Gómez-Navarro on behalf of all co-authors

May 15, 2018

*The paper is well written and addresses a relevant scientific question by describing a promising bias correction method, based on quantile mapping (QM) conditioned to regions with similar temporal variability. It is in general well-structured and represents a substantial contribution to the modelling and impacts community.*

We appreciate the positive view the reviewer expresses about this version of the manuscript. We have tried to address below the concerns he/she poses about the manuscript. We hope the new version improves the deficiencies pointed out by the reviewer.

*Still there are some explanations missing to be able to understand the whole methodology and these explanations may probably answer some of my specific comments. In particular, the regions/clusters are obtained for observations and model independently, I do not understand how the bias correction is trained and applied for each grid box, since the regions are different for each dataset and a grid box may belong to different clusters in both datasets. Thus, how are the calibrated corrections obtained for a region? Which correction is applied to a grid box that belong to different regions in the model and observations?*

As correctly guessed by the reviewer, this is a misunderstanding possibly motivated by insufficient or inaccurate explanations of the details of the methodology. We have carefully edited the manuscript to emphasise these details and clarify how the regions are defined.

In summary, the regions to apply the correction are those defined using the WRF-CESM simulation alone. These regions, i.e. the ones shown in the third column of Fig. 2, are the ones that condition the QM correction. As such, there is no ambiguity in the selection of to which region a given grid point belongs. The reason for the application of this approach is that the aim of the regionalisation is precisely to group regions that (miss)behave similarly, so that we correct them in a similar fashion. The alternative approach, i.e. defining regions based on the observations, would naturally lead to regions that although behave similarly in reality, contain grid points which are in principle affected by biases of different nature in the simulation, which is precisely what the condition of QM to regions tries to avoid.

One could ask, if this misunderstanding comes from including in the discussion the regions obtained in the other datasets, why did include them? We did so because, as stated in the introduction, the aim of this manuscript it twofold. Although the regions in these other datasets are not relevant for the application of bias correction, comparing the regions obtained in WRF-CESM and WRF-ERA, and those within the observations serves for validation purposes. It allows us to draw conclusions regarding the confidence we can put on the ability of these simulations to reproduce the precipitation paterns in the complex area of Switzerland.

*1) If the cluster classification of the raw model data is used, this classification is based on biased data.*

This is indeed the case. The regionalisation is part of the process to remove bias (actually, it is prior to it), therefore it necessarily has to operate on biased data. The regionalisation aims at identifying regions whose precipitation variability is similar, so that we can apply bias correction to such regions coherently. The rationale is that since grid points within a region behave similarly, it makes sense to remove their biases with the same transfer function, because they are arguably affected by the misrepresentation of common physical mechanisms.

A critic of this approach could be that biases might be so prominent that they affect the regionalisation itself, leading to unrealistic/unphysical regions. Fortunately this is not problematic in this case, if this is what the reviewer is arguing. Demonstrating how this is not the case is indeed what the inclusion of the WRF-ERA simulation in this analysis pursues. Figures 2 and 3 compare the results obtained with the regionalisation of WRF-CESM and WRF-ERA. It turns out that, although the former is affected by larger biases (note that WRF-ERA is able to nicely reproduce the annual cycle,

albeit with a consistent wet bias), and the temporal correlation between both runs is negligible (for being CESM a free run), the regions obtained using both datasets present strong similarities. We emphasise this important result in the manuscript:

> In summary, the regions identified in both simulations are similar and resemble the orographical barrier imposed by the Alps. This similarity demonstrates that the spatial structure of precipitation regimes are largely independent on the driving dataset.

*2) If only the classification for the train/test of the QM based on observations is used, how would be the method applied in a changing climate in which the grid boxes could move to another cluster?*

We believe this question is related to the misinterpretation of the methodology addressed in previous answers to this reviewer. The regionalisation is actually based on simulated data, as we have tried to clarify above, so it is not bounded to the existence of observational records, which obviously do not exist under climate change conditions.

However, the approach could be extended to consider this scenario. Under climate change conditions it is in principle perfectly possible to repeat the regionalisation for the biased climate change projection. Once the regions are found, the transfer functions can be obtained for such regions using simulated and observed data for a control period, and then apply the correction to the projected precipitation.

Still, although technically possible, more tricky is to answer to what extent this approach would lead to physical or meaningful results. In the opinion of the authors, the limitations and uncertainties that such a methodology could pose are in line with the raised concerns (now more extensively discussed in the manuscript after the suggestions of this reviewer) about the application of bias correction techniques in a climate change context.

*3) Can one relate those "objective" clusters to e.g. hydrological catchments relevant for impact studies?*

We have tried to relate through the discussion of the results in Section 4.1 the main features of the regions in relation to the most prominent physical characteristics of this area, as they lead to the most clear agreements between the regions found across datasets. Although a more detailed analysis of the precipitation regions, and how they can be related to actual hydrological catchments, could be in principle carried out in this context, it is in our opinion beyond the scope of this manuscript. We believe we should limit ourselves here to the development of the proposed methodology and the validation of two simulations, rather than tackling as well the analysis of the spatio-temporal variability of precipitation and how it relates to hydrological features of Switzerland.

*4) How does the different number of grid boxes in each cluster affect the results? The authors may include the number of grid boxes per region in Fig.2.*

This is an interesting suggestion. We have created a new table (Table 2) with the number of grid points per region. In principle, having fewer grid points leads to transfer functions not so efficiently estimated by the finite sample. However, as the sample consists of pooling all grid points that belong to the same region for the whole period, the number of data points that contribute to the estimation of the quantile-quantile curve that is responsible for the correction is large, i.e. #days within a month in the period 1979-2005 $\times$ #grid points per region. For instance, 48600 pairs of numbers populate the smallest region (Region #6 in WRF-CESM in Winter, with 60 grid points), and this number of much larger in general. We have noted this in the main text (beginning of Section 4.1)

*A further concern is if the authors checked differences/improvements with respect to standard QM (without conditioning to regions). Some discussion about this would be appreciated.*

We have included now a comparison with a simpler method. In particular, we introduce and discuss the results by Felder et al. (2018), being a study which applies a bias-corrected version of the simulations we present here, but carried out with a simpler method. The authors briefly evaluate the model (that study focuses on impacts, and includes several types of models), and the published figures in that reference demonstrate the modest improvement achieved with basic quantile mapping. The study by Felder et al. (2018), which is co-authored by several researchers in the present study, was under review at the time of submitting this manuscript, but it is now published and accessible online, and it can be regarded as the main motivation to develop the new methodology we present here. Therefore, we have included various references to this work in the discussion of the results through the new version of the manuscript.

*Here I list some specific comments and typos, giving the page and line numbers.*
*Specific comments:*
*P1 L6 "minimise disturbances to the physical consistency" -> not clear, please rephrase or elaborate.*

We have rephrased this sentence, as marked in the document that highlight the differences. In the following, we do not discuss in detail the changes, but we invite the editor/reviewer to check such document to evaluate if the changes satisfy the reviewer queries.

*P1 L16 which variables? So far only precipitation was mentioned (also in the title). If the clustering depends on the variable, why does the method preserve the physical consistency among variables more than the standard QM?*

We restrict the sentence to precipitation.

*P2 L3 The authors may consider citing the newer analysis including EURO-CORDEX data: Rajczak, J. and C. Schar (2017), Projections of future precipitation extremes over Europe: a multi-model assessment of climate simulations | J. Geophys. Res. Atmos., doi:10.1002/2017JD027176.*

This reference has been included and commented.

*P3 In the review of bias correction methods, the authors may consider the following paper, with some similarities from a technical point of view, where the bias correction is conditioned to circulation types: Wetterhall, F., Pappenberger, F., He, Y., Freer, J., and Cloke, H. L.: Conditioning model output statistics of regional climate model precipitation on circulation patterns, Nonlin. Processes Geophys., 19, 623-633, https://doi.org/10.5194/npg-19-623-2012, 2012.*

This reference has been included and commented.

*P3 L15 After this paragraph I suggest to include a sentence mentioning the implications in the climate change context, something like "As a consequence, the climate change signal might be unrealistically modified", as stated e.g. by:*
*Casanueva, A., Bedia, J., Herrera, S., Fernández J. and Gutiérrez J.M. Di- rect and component-wise bias correction of multi-variate climate indices: the per- centile adjustment function diagnostic tool. Climatic Change (2018) 147: 411. https://doi.org/10.1007/s10584-018-2167-5 Teng J, Potter NJ, Chiew FHS, Zhang L, Wang B, Vaze J, Evans JP (2015) How does bias correction of regional climate model precipitation affect modelled runoff? Hydrol Earth Syst Sci 19(2):711–728. https://doi.org/10.5194/hess-19-711-2015*

Such sentence and references have been included.

*P3 L16-20 The authors may consider the above paper (Rajczak and Schar 2017) to update that summary of previous works.*

This reference has been included.

*P3 L20-21 what do the authors mean with "similar"? different model version? Parameterizations?*

We refer to what the authors show in this publication: same model configuration, just different spatial resolution. We have edited the text accordingly.

*P3 L16-25 I would suggest to move the entire paragraph before the previous one, in which bias correction is introduced, since it reads better after line 3 and here it is again about previous studies in which bias correction is not applied. Also the final lines of the paragraph (23-25) are more or less repeating what it is already said in P2 L34.*

We agree with this suggestion, so we have swapped the paragraphs.

*P5 L24 Is there a reason for using 27 years instead of e.g. 30?*

The reason is availability of global data to drive the RCM simulations. The ERA-Interim and CESM runs span different periods of time, but for comparison and validation purposes we focus the analysis on the overlap of both datasets. The ERA-Interim period starts in 1979, therefore this provides the lower bound. The CESM simulation runs up to 2005 (starting in 1850), which provides the upper boundary. We have carefully rephrased this in section 2.5 to explain this detail. The WRF-ERA simulation is actually longer, but we did not include the 2006-2013 period for consistency with WRF-CESM.

*P7 L7 I suggest to add "smooths out the transfer functions prior to the correction".*

We modified the text accordingly.

*P7 L9 Until now it is not clear which is the analysis domain, the title says Alpine region, simulations are performed for the Alpine region but observations are available for Switzerland. Please consider to mention Switzerland explicitly in the experimental design and title.*

We acknowledge that the difference between the area that is downscaled (the entire Alpine area) and the one we analyse in detail due to available observations (Switzerland) was not clear enough in the first version of the manuscript. Therefore, we have emphasised this difference in the new version in sections 1, 2.4 and 6.

Regarding the title, we believe that including "Switzerland" in it seems to artificially limit the scope of our analysis, as both the simulations we present here, as well as the bias correction techniques, are not limited to this country. Therefore, we have opted for *"A new region-aware bias correction method for simulated precipitation in areas of complex orography"*

*P7 L16 As mentioned before, the authors mention the preservation of physical consistency. My question now is how coherent is the method in a multivariate case? I guess a different division in clusters would be performed for each variable. Can the authors comment something on this?*

We have adjusted the sentence to limit the scope of what physical consistency is meant here. As we are dealing with a single variable, breaking the physical consistency would imply in this context that we correct the precipitation in a way that it breaks down the spatial and temporal structure of variability of this single variable. For instance, corrections could smooth out differences in precipitation in opposite sides of a mountain, therefore destroying part of the added value of the high-resolution simulation, or even breaking conservation laws (of mass, in this case) implicit in the simulation.

*P7 L21 "varies per season" why seasonally? In section 3 it is said that the method is applied for each month, thus one expect to have different clusters at the monthly scale.*

The reviewer is right, and this is a detail that was not very clearly explained in the first version of the manuscript. The correction is indeed applied to each month separately to efficiently account for the annual cycle. Therefore, in principle the regionalisation, which is a prerequisite for obtaining the transfer functions that conform the correction, should be carried out on a monthly basis as well. However this has some drawbacks. First, the computational cost of the clustering is relatively high: $3 \times 12$ regionalisations should be carried out, each of which including a previous EOF analysis and the clustering of the resulting temporal series of principal components. Second, it leads to 36 maps that we should show and analyse. But eventually, the details that these 36 maps provide are limited, as there exist great redundancy across the annual cycle, because months that belong to the same season behave similarly in terms of precipitation. Having this into account, we decided to apply a simplification that consists of performing the regionalisation on a seasonal, rather than a monthly basis, and then using the same regions for the three months within each season. This simplification does not impact the outcome of the correction, while it optimises the computational cost and reduces by a factor of 3 number of maps needed to show and have into consideration to discuss the similarities and differences among datasets and through the annual cycle that is supported by Fig. 2. Therefore, we have added a whole new paragraph to Section 3:

> We note that the application of this methodology implies a previous regionalisation of the series for each month separately, which in general involves notable computational cost. Further, months belonging to the same season behave similarly, so that the resulting regions are hardly distinguishable and the analysis presents some level of redundancy. For these reasons, we propose a simplified form of the methodology, which we apply hereafter, and consists of carrying out the regionalisation on a seasonal basis. Once identified, these regions can be regarded as representative and common for the three months within each season, so that the final correction can be applied on a monthly basis.

*P8 L13 The authors mention several times the insufficient effective resolution of the observations, what about the effective resolution of the simulations? The authors should include it in the discussion as well.*

Perhaps "insufficient" is not an appropriate term here, as it suggest an absolute measure. In this part of the manuscript we compare the results obtained from the WRF simulations with the observational product, and try to attribute the differences to physical mechanisms. The difference in the spatial

resolution is an obvious candidate, as the effective resolution of the gridded product acknowledged by its authors is about one order of magnitude lower than the one implemented in both simulations. The "effective resolution" in RCM simulations is difficult to stablish, but it is generally accepted to be between 2 and 4 times larger than the spatial resolution, depending on the variable (Pielke Sr, 2013). Therefore, the spatial resolution of the gridded product is lower than the one in the simulations, and precludes it from capturing the finer orographic features of the domain of study, especially over mountain tips, which in turn can be resolved to a generally larger extent by the simulations presented in the manuscript. Therefore, a word that suggest the comparison of relative measures is perhaps more adequate. We have rephrased this sentence and included the comment about effective resolution of RCMs accordingly.

*P8 L15-35 The authors should motivate better the correlation analysis in Fig.3. I do not see the point of this analysis, especially since the clusters are built in a way that the differences among clusters are maximized. Moreover, the clusters are different in each dataset, so there is not a clear correspondence. This lack of correspondence is only mentioned and resolved in Fig.4.*

The aim of the correlation analysis is to provide a quantitative assessment of the level of similarity/dissimilarity among regions. Note that the regionalisation step will always produce a number of regions, but how different these regions really are is something that can not be answered by looking at Fig. 2 alone. The correlation analysis shown in Fig. 3 tries to overcome this caveat by providing numbers that allow to better judge objectively the coherence of the regions.
We have rephrased this introductory paragraph to motivate better this analysis.

*P9 L31 "averaged over Switzerland" Given the differences in the annual cycle among the regions, the authors may consider doing this analysis per cluster, based on the observations or the WRF-ERA classifications.*

Although we believe the reviewer makes here a valuable suggestion, we have been carefully considering how to account for it. Finally we have decided to leave the figure as is for a number of reasons, including:

- It is not clear which regions we should use. Figure 2 shows how each dataset leads to three possibilities, and they are all in principle equally valid for such an analysis. Should we consider and show all possible variations?

- For each dataset, the regions vary through the annual cycle. Should we change the way the average is calculated through the annual cycle? This would complicate the way the results have to be read.

- Considering all possibilities and showning them, we would end up with a very complex figure with tens of bars which would make the reading and interpretation of the figure difficult.

- The aim of this figure is to illustrate, in general terms, how precipitation varies through the annual cycle across datasets. The point of the figure is to show the consistent overestimation of precipitation in WRF-ERA, as well as the seasonality isues of WRF-CESM. For a spatially disaggregated version, that illustrates the different behaviour across the domain, the reader can get further insight in Figure 6.

- We believe it is important achieve a certain level of consistency with similar studies that facilitates their inter-comparability. In this regard, Figures 5 and 6 are, in their current shape, easily comparable to similar results in other references suggested by the reviewers, as indeed we do in the new version of the manuscript (e.g. Torma Csaba et al.; Fantini et al.). Changing this would dificult or make imposible such a comparison.

*P10 L1-8 The underestimation of precipitation in the Ticcino during autumn is worth to mention. Can the authors give a reason for this?*

We have discussed this issue through personal communication with other researchers working with WRF in the same target region,. They have also found a negative precipitation bias over Ticino in summer, and they traced it to a negative moisture bias in the lowest layers. However, to thoroughly test this, we would need to step by step check if WRF represents all processes detailed below correctly, in a comprehensive analysis of this bias and its causes which is beyond the scope of the manuscript. Still, we have included a brief discussion of the possible causes in the main text.

Isotta et al. (2014) show that in the region of Ticino up to 70% of the yearly precipitation accumulation is due to the top 25% of the wet days, so it is sensible to assume that the bias stems from high to extreme precipitation events. In Ticino these heavy precipitation events are driven by the transport of moist and potentially unstable (moist neutral stratification) air masses against the Alps from the south (Martius et al., 2006; Froidevaux and Martius, 2016). Locally, the vertical shear between south-easterly flow near the surface and southerly to southwesterly above 850 hPa leads to moisture convergence and repeated formation convective cells (Panziera et al., 2015). On an even more local scale, strong vertical shear can result in small-scale circulation that results in local precipitation maxima (Houze et al., 2001). Therefore if the RCM fails to capture any of these local and highly driven by the orography processes properly, it will result in an underestimation of the precipitation.

*P10 L12 The authors should also explicitly mention in the methods how the precipitation frequency is adjusted by this method (relevant for the interpretation of Fig.7). Standard QM is able to correct for a higher frequency of wet days in the model, but the opposite problem (here shown in Fig. 7, winter) could be corrected by applying the frequency adaptation, otherwise an overestimation of the wet day frequency is found in the corrected data. See : Themeßl, M.J., Gobiet, A. & Heinrich, G. Empirical-statistical downscaling and error correction of regional climate models and its impact on the climate change signal. Climatic Change (2012) 112: 449. https://doi.org/10.1007/s10584-011-0224-4*

We do not use frequency adaptation techniques. This indeed leads to the wet bias in the corrected precipitation in winter, as pointed out by the reviewer. We now explicitly acknowledge this limitation and point out how it could be a suitable solution to this bias.

*P12 L4 Why are the temporal correlations lower in autumn? This may be related to the way the corrections are trained and applied.*

This is motivated by the variability of biases within this season, and is related to the cancellation of biases in the intermediate seasons further discussed in response to reviewer #2 in the context of Fig. 7. The figure below shows the dispersion map of the raw versus corrected series in a single grid point where correlation is low (i.e. a grid point within one of the red spots in Fig. 8). For each season, the points line up around three different curves, which are the transfer functions for each of the months within the season. The large spread Autumn (orange) is evident, especially when compared to the homogeneity of the three hardly distinguishable curves for winter (blue) and summer (red). This spread deteriorates the linear relationship between raw and corrected data, and therefore reduces the correlation. Although we do not show this figure in the manuscript for the sake of brevity, we have introduced a short explanation of this effect in the main text.

[Figure]

*P22 Fig.4 The decimal dots are missing in the labels of the Taylor diagram. And more important than that, it is completely unclear to me what is shown by the angular scale (azimuthal angle). I would expect to have represented there correlation values but that legend must be something else. Please explain in the caption how this should be interpreted.*

This is a standard Taylor diagram, where the angular scale represents correlation. The missing points (and labels!) were not missing in our original manuscript. Unfortunately some technical issue in the conversion to produce final uploaded document seems to have removed some of the information in the figure. We include below the figure as it was supposed to be included in the manuscript.

[Figure]

We believe this figure does not lead to the issues raised by the reviewer, and certainly we will be more careful in the final submission.

*Technical corrections:*

All technical corrections have been implemented as suggested by the reviewer.

**References**

Fantini, A., Raffaele, F., Torma, C., Bacer, S., Coppola, E., Giorgi, F., Ahrens, B., Dubois, C., Sanchez, E., and Verdecchia, M.: Assessment of multiple daily precipitation statistics in ERA-Interim driven Med-CORDEX and EURO-CORDEX experiments against high resolution observations, pp. 1–24, https://doi.org/10.1007/s00382-016-3453-4.

Felder, G., Gómez-Navarro, J. J., Zischg, A., Raible, C. C., Röthlisberger, V., Bozhinova, D., Martius, O., and Weingartner, R.: From global circulation to local flood loss: Coupling models across the scales, Science of the Total Environment, 635, 1225–1239, https://doi.org/10.1016/j.scitotenv.2018.04.170, 2018.

Froidevaux, P. and Martius, O.: Exceptional integrated vapour transport toward orography: an important precursor to severe floods in Switzerland: Integrated Vapour Transport and Floods in Switzerland, 142, 1997–2012, https://doi.org/10.1002/qj.2793, 2016.

Houze, R. A., James, C. N., and Medina, S.: Radar observations of precipitation and airflow on the Mediterranean side of the Alps: Autumn 1998 and 1999, 127, 2537–2558, https://doi.org/10.1002/qj.49712757804, 2001.

Isotta, F. A., Frei, C., Weilguni, V., Perčec Tadić, M., Lassègues, P., Rudolf, B., Pavan, V., Cacciamani, C., Antolini, G., Ratto, S. M., Munari, M., Micheletti, S., Bonati, V., Lussana, C., Ronchi, C., Panettieri, E., Marigo, G., and Vertačnik, G.: The climate of daily precipitation in the Alps: development and analysis of a high-resolution grid dataset from pan-Alpine rain-gauge data: CLIMATE OF DAILY PRECIPITATION IN THE ALPS, 34, 1657–1675, https://doi.org/10.1002/joc.3794, 2014.

Martius, O., Zenklusen, E., Schwierz, C., and Davies, H. C.: Episodes of alpine heavy precipitation with an overlying elongated stratospheric intrusion: a climatology, 26, 1149–1164, https://doi.org/10.1002/joc.1295, URL http://doi.wiley.com/10.1002/joc.1295, 2006.

Panziera, L., James, C. N., and Germann, U.: Mesoscale organization and structure of orographic precipitation producing flash floods in the Lago Maggiore region: Orographic Convection in the Lago Maggiore Area, 141, 224–248, https://doi.org/10.1002/qj.2351, 2015.

Pielke Sr, R. A.: Mesoscale meteorological modeling, vol. 98, Academic press, 2013.

Torma Csaba, Giorgi Filippo, and Coppola Erika: Added value of regional climate modeling over areas characterized by complex terrain—Precipitation over the Alps, 120, 3957–3972, https://doi.org/10.1002/2014JD022781.

---

## Author Comment (AC2) · 15 May 2018

Thank you very much for the detailed review and very constructive comments. We have reviewed them and we think we can address them in a new version we are currently preparing. I attach to this comment a PDF document we have prepared with a point-by-point response to all the queries.

Please also note the supplement to this comment: https://www.geosci-model-dev-discuss.net/gmd-2017-329/gmd-2017-329-AC2-supplement.pdf

[Figure]

[Figure]

**Supplement:**

**Point-by-point response to Reviewer #2**

J. J. Gómez-Navarro on behalf of all co-authors

May 15, 2018

**Anonymous Reviewer #2:**

*This paper presents a bias correction method for regional climate simulations over the Alps at very high resolution. A observational database for the region is used for the validation, and ERAinterim and GCM-CESM forcing fields are used to WRF modelling work. To my opinion, it shows enough aspects to novelty and adequate analysis and understanding of the obtained results. I suggest it to be considered for publication, once the questions and requested item can be properly answered or at least taken into account in some way*

We thank the reviewer for the time devoted to carefully read the manuscript, the positive vision expressed about it, and the constructive comments that will certainly improve the final version. We have tried to answer point by point all his/her comments below.

*1. Missing references. It is always the case that not all the relevant references are included when a work is presented. Here I find some that I consider that are essential to be included, not only for the introductory aspects, but also for the methods and results description. Let me indicate them to the authors for them to be considered a properly used throughout the text*

The new version of the manuscripts includes many more references, including most of those suggested by both reviewers, and that clearly allow to better contextualize this piece of work in the existing literature.

*2. Apart from the pure bibliography missing items, there are some aspects that could be more deeply described by the authors. One of them should be to compare the proposed bias correction method with other similar ones, if there are some, to see more clearly differences and similarities with others already proposed. I am sure the quantile mapping procedures have been used before, if one goes to those references. Therefore, I recommend the ongoing work by Nikulin and others in the frame of EuroCORDEX activities, named BCIP. Take a look at this abstract at EGU2015: Nikulin, G., Bosshard, T., Yang, W., Bärring, L., Wilcke, R., Vrac, M., ... & Fernández, J. (2015, April). Bias Correction Intercomparison Project (BCIP): an introduction and the first results. In EGU General Assembly Conference Abstracts (Vol. 17). In a more general sense, perhaps a mention to this recommendation by CORDEX community could be made. take a look at http://cordex.org/data-access/bias-adjusted-rcm-data/, and from there, to a IPCC work focused on this topic: See Breakout Group 3bis: Bias Correction (pp. 21-23) in IPCC, 2015: Workshop Report of the Intergovernmental Panel on Climate Change Workshop on Regional Climate Projections and their Use in Impacts and Risk Analysis Studies [Stocker, T.F., D. Qin, G.-K. Plattner, and M. Tignor (eds.)]. IPCC Working Group I Technical Support Unit, University of Bern, Bern, Switzerland, pp. 171. (https://www.ipcc.ch/pdf/supporting-material/RPW_WorkshopReport.pdf). I can imagine that authors do not want to go too far on this aspect, but I think that some more comments, to have this work inside the wider context, should be made. Even a mention to some developed software for this kind of analysis could be included, such as Bedia, J., Iturbide, M., Herrera, S., Manzanas, R., & Gutiérrez, J. (2017). downscaleR: an R Package for Bias Correction and Statistical Downscaling. R Package Version 2.0-3.*

We have included a discussion of the issues raised by the reviewer about on-going debate about the use of bias correction, including the mentioned references in the new version of the manuscript.

*3. I am not sure if the authors have a comment about the fact that this bias correction method has been applied to a region with a very deep orography, and to precipitation field. Which could be the potential to apply it to other regions with smoother orography, and/or to other variables?*

We have added a new paragraph in the conclusions to discuss how this method can be exported to other regions/variables. We reproduce it below:

We note that the rationale of the developed methodology is to divide a large domain into smaller subregions according to the behaviour of the target variable. We have applied it here to daily precipitation in Switzerland for being a variable strongly affected by complex orographical details that lead to strong horizontal gradients. With more generality, spatial regionalisation is an efficient method to break down complexity in areas and variables whose behaviour strongly varies through the domain. Still, the bias correction applied separately to subregions can be in principle adapted to other cases with simpler topography, or other variables with lower horizontal gradients. The only practical difference is that in this case the regionalisation will naturally lead to a lower number of subregions which are necessary to obtains clusters with coherent features.

*Specific comments*

*1. It has been indirectly mentioned on the general comments section, but here I want to comment if explicitly: I miss a mention to the EuroCORDEX/MedCORDEX activities, that have used plenty of simulations at high resolutions (0.11) over Europe, and several studies with not a single RCM as here, but an ensemble of them, that have analyzed, also forced with ERAinterim fields, how precipitation is described. I do not mean a full comparison with other RCMs, but at least some mention and comparison with them, to see more clearly if WRF-RCM is similar to the state-of-the-art RCMs modelling alpine precipitation for current climate conditions.*

We have added plenty of references and explicit mentions to EURO-CORDEX and MED-CORDEX activities in the newer version.

*2. And also related to this point, I miss some comparison of your figure 5, for example, with figure 2 of Torma et al., 2015 or Fantini et al., 2016, figure 5, not only for RCMs, but also for observational datasets, I am not sure if they are totally consistent. Or for your figure 6 and 7, and their corresponding figures.*

This is an important pitfall that has been corrected in the new version. We have enlarged the discussion of the results in Section 4.2 with explicit mention to the ones in the corresponding figures in the two sources pointed out by the reviewer. However, we have not included a discussion comparing the results about daily PDFs in Fig. 7. The reason is that the daily PDFs in those references include all seasons, and they are built to emphasise the different results across spatial resolutions. Therefore they are somewhat different figures, and it is difficult to stablish a fair and meaningful comparison.

*3. I have a concern about the domain of study chosen here. On figure 1, D4 subdomain seems to be the one used for the analysis, but then figures with the political borders of Switzerland seem to be used. This relatively artificial borders could add some non-physical or modelling aspects to the analysis, and specially when obtaining the subregions from the clustering procedure. Which is the opinion of the authors about this aspect?.*

This confusion between the simulated and analysed domains (the Alpine region vs. Switzerland) has also been pointed out by reviewer #1. We believe we were not clear enough in the former version in the description of the dataset and the methodology. Therefore we have clarified this in the newer version of the manuscript. The reason for using Switzerland, i.e. a political boundary, instead of a natural one, is that the observational product we used is limited to this domain. This imposes a unavoidable bottleneck of the validation. Certainly there are observations beyond the borders of Switzerland, but we believe that they do not contain the high density, even distribution, and quality-tested of the observations blended to create the gridded product developed by Meteoswiss.

*4. Another point I would like to hear from the authors is about the very high resolution used for the WRF D4 domain (page 5, line 14): 2km. Which one is the real advantage here of using such resolution compared with the even-very-high 6km one?. It seems that no much mention or usefulness is made by the authors to this resolution, by far much larger than the mentioned 0.11 "high resolution" EuroCORDEX standard values these days. It is also a tricky aspect, since the comparison and bias correction method is made against the roughly 20km observational dataset information, and so some statements are made through the text related to this resolution differences. A more complete study should perhaps include at least some other resolution from the WRF model to a better understanding of the resolution topic?.*

We have tried to motivate the added value of the high resolution. In particular, the fact that we can avoid the use of parametrisations of convective processes, and we provide references that back the

added value of such simulations. The difference between 2 km and 6 km can be substantial. For instance, in response to reviewer #1 we have discussed the effective resolution. If we use the factor 3-4 mentioned in some references (e.g. Pielke Sr, 2013), 6 km of spatial resolution would have an effective resolution clearly above 10 km, which can be argued that it is not sufficient to account for all convective processes. This can be hardly put in doubt with a resolution of 2 km. More precisely, (Gómez-Navarro et al., 2015) investigated the particular issue of the skill as a function of spatial resolution, and found that there is indeed a large gain in switching from 6 km to 2 km. Unfortunately the latter study is based on the performance of surface wind, not precipitation. With this context, it seems reasonable to carry out a study of the added value of the model performance as a function of spatial resolution, using the precipitation produced within D3 or even D2 of this simulation. Unfortunately such an analysis can not be carried out with the present simulation. The reason is that this run was carried out with all domains nested two-way, as described in section 2.4. This implies that the precipitation as simulated by each coarser domain is replaced by the one within the innermost domain in the overlap region, i.e. the precipitation recorded for D3 inside the region span by D4 is actually a spatial smoothed version of the latter. Therefore it is does not correspond to the actual precipitation as resolved by a 6 km configuration, but an improved version that accounts for phenomena explicitly resolved within D4. This effectively precludes the use of this data for the fair evaluation of the model performance as a function of spatial resolution suggested by the reviewer. At this stage, a proper evaluation of this issue would require re-running great part of the simulations, which would involve a prohibitive computational cost.

*5. I understand that the forcing GCM is always an open question, but the usage of just one instead of, at least, a couple of them, does not limit a little bit the representativity of the GCM-forced RCM analysis?*

Certainly. It is always better to target at an ensemble, as such an approach allows to better characterise GCM-specific biases. This is indeed what we aim to some extent with the inclusion of the simulation driven by ERA-Interim in the analysis. However, computational cost is a bottleneck in this study. Carrying out a single realisation with a single GCM costed thousand of hours in one of the most powerful supercomputer available, CSCS. It is completely unaffordable for us the repetition of this simulation driven by alternative GCMs to produce a proper ensemble. We hope that this limitation is overcome in future studies, but unfortunately we are currently limited by this.
Still, we have added a paragraph in section 2.5 to discuss this issue.

*6. The result shown in pages 10-11 that related intermediate seasons with cancellation artifacts sounds reasonable, but perhaps a more specific analysis could be made, with moving seasons, to see if more clear picture of that can be obtained. Because on the other hand, this result could be found non-intuitive, as one can think that precisely those transition seasons are more difficult to be properly captured. Which are the thoughts of the authors about it?.*

The figure below shows the result of the calculation suggested by the reviewer. It shows PDFs of daily precipitation within "moving seasons". There are 12 panels, each one obtained considering as the window the given month, the previous and the former. The coloured panels highlight the standard seasons shown in Fig. 7 in the manuscript. The compensation of errors in intermediate seasons becomes apparent in WRF-CESM, as this simulation shows opposite biases in the previous and following seasons. We have briefly discussed this results in the manuscript, although we believe that the inclusion of the figure is not necessary.

[Figure]

7. Page 11, line 22. The bias corrected result over the frequency distribution that changes from underestimation to overestimation in winter looks a little bit peculiar. Could this result be a little bit further explained?

This issues has been raised by reviewer #1. As discussed by Themeßl et al. (2011), this effect occurs when models tend to underestimate the dry-day frequency (which is a rather infrequent feature of some RCMs, as most of them exhibit the oposite behaviour, i.e. drizzling-effect), as all these days become mapped onto a precipitation day, leading to a wet bias. This could be further corrected using frequency adaptation techniques, although we have not considered such techniques here. A brief discussion of this aspect has been included in the manuscript.

*Technical corrections*

1. When describing the experimental design (page 5, line 25) I do not understand those 6-day chunks and 12h spinup periods. I thought that a whole year or even two or more where needed for the soil moisture to be adapted. Could this aspect be explained a little bit more? I understand that more details can be found in Gomez-Navarro et al., 2015, but perhaps here it is too little what is said. It is the same about D1-D2-D3-D4 subdomains and nesting aspects.

We carry out the simulation in so-called reforecast mode. This approach is not new, but a well-settled methodology to conduct RCM simulations that splits the simulations into small tranches. As explained in the cited reference (Gómez-Navarro et al. 2015), *"The method consists of splitting a long simulation into shorter simulation periods of 1 to a few days, running each period separately and finally merging them. This method effectively minimises the impact of the boundaries, transforming the problem into a mostly initial-value problem. The reforecast method is regularly applied (Jiménez and Dudhia, 2012; García-Díez et al., 2013; Menendez et al., 2014, among others), and the increased skill of this method compared to continuous runs has been reported (Lo et al., 2008)."*. In a nutshell, splitting the simulation allows to bind the RCM to the driving dataset, and it can

be regarded as a form of nudging. Further, this strategy has computational advantages: several simulations can be run simultaneously, which naturally leads to the parallelization of the problem.

Of course there are drawbacks. As pointed out by the reviewer, the short spinup period does not allow the soil moisture to reach an actual equilibrium with the atmosphere, which in opinion of the authors reduces the land-atmosphere coupling (still, this coupling does not disappears, as it is borrowed from both the soil and atmosphere initial conditions used to run both submodels within each tranche). Although we believe this can bias certain applications of the simulations, for instance in the study of severe droughts and certain type of floodings, this approach does not impose a fundamental problem in general, as the successful validation of the simulations carried out in this same study demonstrates.

Regarding the domains, we clearly state their setup in Section 2.4 and even show them explicitly in Fig. 1: "Horizontally, we use four two-way nested domains with grid sizes of 54, 18, 6 and 2 km, respectively (top map in Fig. 1)"

We have added more details and a brief exposition of these arguments in the Section 2.5 in new version of the manuscript.

*2. Close to this point, I do not also understand why nudging is applied to ERAinterim forced simulation, but not to the ESM one.*

This is not arbitrary, but there is a rationale behind this choice. We developed it in the first version of the manuscript:

> The rationale behind this choice is that avoiding nudging gives the model more freedom to develop a more precise representation of the physical processes at regional scales (due to the higher resolution), and thus is potentially able to better correct systematic biases of the ESM, which, e.g., simulate a too strong zonal circulation (Bracegirdle et al., 2013)."

**References**

Gómez-Navarro, J. J., Raible, C. C., and Dierer, S.: Sensitivity of the WRF model to PBL parametrisations and nesting techniques: evaluation of wind storms over complex terrain, Geoscientific Model Development, 8, 3349–3363, https://doi.org/10.5194/gmd-8-3349-2015, 2015.

Pielke Sr, R. A.: Mesoscale meteorological modeling, vol. 98, Academic press, 2013.

Themeßl, M. J., Gobiet, A., and Leuprecht, A.: Empirical-statistical downscaling and error correction of daily precipitation from regional climate models, International Journal of Climatology, 31, 1530–1544, https://doi.org/10.1002/joc.2168, 2011.